# Biomolecular condensates mediate bending and scission of endosome membranes

Yanning Wang[1,14], Shulin Li[1,14], Marcel Mokbel[2,14], Alexander I. May[3,4], Zizhen Liang[5], Yonglun Zeng[6,7], Weiqi Wang[5], Honghong Zhang[1], Feifei Yu[8], Katharina Sporbeck[3], Liwen Jiang[5], Sebastian Aland[2,9], Jaime Agudo-Canalejo[10,11], Roland L. Knorr[3,12,13 ✉] & Xiaofeng Fang[1 ✉]

Multivesicular bodies are key endosomal compartments implicated in cellular quality control through their degradation of membrane-bound cargo proteins[1–3]. The ATP-consuming ESCRT protein machinery mediates the capture and engulfment of membrane-bound cargo proteins through invagination and scission of multivesicular-body membranes to form intraluminal vesicles[4,5]. Here we report that the plant ESCRT component FREE1[6] forms liquid-like condensates that associate with membranes to drive intraluminal vesicle formation. We use a minimal physical model, reconstitution experiments and in silico simulations to identify the dynamics of this process and describe intermediate morphologies of nascent intraluminal vesicles. Furthermore, we find that condensate-wetting-induced line tension forces and membrane asymmetries are sufficient to mediate scission of the membrane neck without the ESCRT protein machinery or ATP consumption. Genetic manipulation of the ESCRT pathway in several eukaryotes provides additional evidence for condensate-mediated membrane scission in vivo. We find that the interplay between condensate and machinery-mediated scission mechanisms is indispensable for osmotic stress tolerance in plants. We propose that condensate-mediated scission represents a previously undescribed scission mechanism that depends on the physicomolecular properties of the condensate and is involved in a range of trafficking processes. More generally, FREE1 condensate-mediated membrane scission in multivesicular-body biogenesis highlights the fundamental role of wetting in intracellular dynamics and organization.

Biomolecular condensation has recently emerged as a key mechanism of intracellular compartmentalization that has a range of intricate roles in cellular organization[7,8]. Condensates form by liquid–liquid phase separation (LLPS) of key scaffold molecules, with various client molecules incorporated into the condensate by partitioning[9]. Condensate physical properties are determined by both intracondensate cohesive forces (surface tension) and condensate–substrate adhesive (wetting) forces that give rise to capillarity, which is responsible for evolutionarily conserved physiological functions[10,11]. For example, capillary forces mediate the rapid remodelling of diverse membrane-bound organelles implicated in dynamic processes, such as the formation of autophagosomes[12] and protein storage vacuoles[13]. However, how such complex membrane morphological changes occur, including terminal membrane scission events, is not well understood.

The ESCRT machinery, comprising subcomplexes 0, I, II and III, is a class of highly conserved proteins involved in membrane remodelling, including the formation of neck-like membrane invaginations and subsequent neck scission events. The biogenesis of multivesicular bodies (MVBs) is well established as an ESCRT-mediated remodelling process[1]. During MVB generation, ESCRTs orchestrate the sorting of ubiquitinated membrane cargoes into membrane invaginations that ultimately form intraluminal vesicles (ILVs) inside MVBs. ILV scission from MVB membranes is achieved through the ATP-consuming constriction of membrane-bound filamentous ESCRT spirals[2,4]; these ILVs are degraded after MVB fusion with a lytic compartment such as the lysosome. ESCRTs also contribute to retroviral budding at the plasma membrane, autophagosome sealing and nuclear repair[3,5]. In mammals, ESCRT perturbation has been linked to neurodegenerative diseases and

[1]School of Life Sciences, Tsinghua University, Beijing, China. [2]Faculty of Mathematics and Informatics, Technical University Freiberg, Freiberg, Germany. [3]Institute of Biology, Faculty of Life Sciences, Humboldt-Universität zu Berlin, Berlin, Germany. [4]Cell Biology Center, Institute of Innovative Research, Tokyo Institute of Technology, Yokohama, Japan. [5]Centre for Cell and Developmental Biology and State Key Laboratory of Agrobiotechnology, School of Life Sciences, The Chinese University of Hong Kong, Hong Kong, China. [6]State Key Laboratory of Plant Diversity and Specialty Crops and Guangdong Provincial Key Laboratory of Applied Botany, South China Botanical Garden, Chinese Academy of Sciences, Guangzhou, China. [7]University of Chinese Academy of Sciences, Beijing, China. [8]College of Grassland Science and Technology, China Agricultural University, Beijing, China. [9]Center for Systems Biology Dresden, Dresden, Germany. [10]Max Planck Institute for Dynamics and Self-Organization, Göttingen, Germany. [11]Department of Physics and Astronomy, University College London, London, UK. [12]Graduate School and Faculty of Medicine, The University of Tokyo, Tokyo, Japan. [13]University of Cologne, Faculty of Medicine and University Hospital Cologne, Cologne, Germany. [14]These authors contributed equally: Yanning Wang, Shulin Li, Marcel Mokbel. ✉e-mail: roland.knorr@uni-koeln.de; xffang@tsinghua.edu.cn

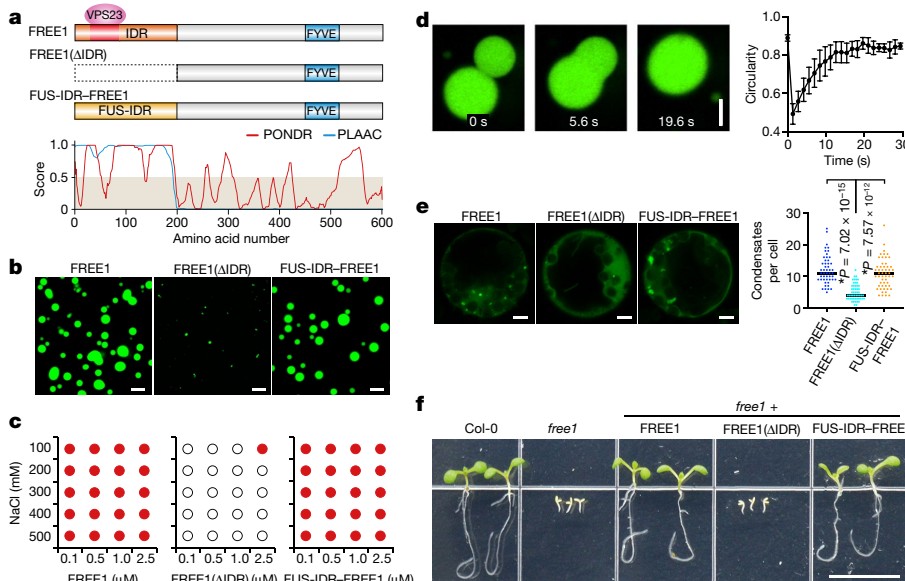

**Fig. 1 | FREE1 phase separates in vitro and forms functional condensates in vivo. a**, The protein domain structures of FREE1 and its variants (top). Bottom, intrinsically disordered and prion-like regions were predicted using the PONDR and PLAAC algorithms. **b**, In vitro phase-separation assay of 2.5 μM GFP-labelled FREE1 variants. **c**, Phase diagrams of FREE1 variants. The red dots indicate LLPS. The empty circles indicate no LLPS. **d**, Coalescing FREE1 condensates. Left, time-course imaging. Right, circularity of fusing condensates over time. The final diameter of condensates is 10–14 μm. Data are mean ± s.d. $n$ = 7 condensates.

**e**, FREE1 condensates in vivo. Left, representative images of *A. thaliana* protoplasts expressing GFP-labelled FREE1 variants. Right, quantification of the condensate number per cell. Black lines indicate the median. $P$ values were calculated using two-tailed $t$-tests. $n$ = 50, 53 and 53 protoplasts, respectively. The asterisks indicate significant differences. **f**, Seven-day-old *A. thaliana* seedlings of the indicated genotypes. Representative images of $n$ = 3 (**b**, **d** and **e**) and $n$ = 2 (**f**) independent experiments. Scale bars, 5 μm (**b**, **d** and **e**) and 1 cm (**f**).

cancer[14,15]. In plants, loss of ESCRT results in seedling lethality[16] for yet undetermined reasons. Here we find that condensation of the plant ESCRT accessory protein FREE1 can mediate MVB formation. Our results establish an ESCRT-machinery-independent mechanism of membrane scission by forces that originate in condensate–membrane wetting.

## FREE1 forms functional condensates

The small molecule biotinylated isoxazole (b-isox) has been used to precipitate phase-separating proteins from plant lysates[17]. After b-isox treatment (Methods), we detected enrichment of FREE1 in the precipitate by three orders of magnitude, as well as less-marked enrichment of several other ESCRT complex proteins (Extended Data Fig. 1a,b). The presence of intrinsically disordered regions (IDRs) within proteins is associated with phase-separation behaviour[7]. Although most ESCRT complex proteins identified using b-isox include extended IDRs (Fig. 1a and Extended Data Fig. 1a,c), we found that only purified FREE1 phase separates in vitro (Fig. 1b and Extended Data Fig. 1d) in a manner that is unaffected by salinity and its fusion with a GFP moiety (Fig. 1c and Extended Data Fig. 2a,b). We confirmed that FREE1 condensates coalesce (Fig. 1d and Extended Data Fig. 2c), and fluorescence recovery after photobleaching (FRAP) showed recovery of fluorescence signal (Extended Data Fig. 2d), indicating liquid-like material properties. FREE1 condensate formation was prevented by deletion of the FREE1 IDR (FREE1(ΔIDR)) (Fig. 1a–c) or addition of a solubilizing MBP tag (Extended Data Fig. 2e).

We further found that transient expression of FREE1 in *Arabidopsis thaliana* protoplasts (Fig. 1e) resulted in the formation of condensate-like structures that were sensitive to the condensate-dissolving reagent 1,6-hexanediol[18] (Extended Data Fig. 2f). By contrast, expression of FREE1(ΔIDR) did not lead to condensate formation (Fig. 1e). Heterologous expression of these proteins in mammalian cells also resulted in IDR-dependent condensate formation (Extended Data Fig. 2g), further demonstrating that FREE1 is a phase-separating

protein, the IDR of which is required for condensate formation. We next confirmed that genetic knockout of *FREE1* (*free1*) results in seedling lethality, as reported previously[6,19]. We rescued this phenotype by expressing condensate-forming FREE1 but not the non-condensing FREE1(ΔIDR) variant (Fig. 1f and Extended Data Fig. 3), suggesting that FREE1 condensate formation is essential for its function. To further test this hypothesis, we constructed FREE1 chimeras in which the IDR of FREE1 was replaced by heterologous IDRs from either the fused in sarcoma (FUS)[20–22] or FLOE1[23] proteins. While IDRs of FUS and FLOE1 are physiologically unrelated to the IDR of FREE1, all three IDRs are prion-like domains of similar length and comparable proportions of amino acid classes (Extended Data Fig. 2h,i). As for wild-type FREE1, we observed that FUS-IDR–FREE1 and FLOE1-IDR–FREE1 chimeras phase separate in vitro (Fig. 1b,c, and Extended Data Fig. 2j,k) and after expression in protoplasts (Fig. 1e). Importantly, both FUS-IDR–FREE1 and FLOE1-IDR–FREE1 rescued *free1* seedling lethality (Fig. 1f and Extended Data Fig. 3a–e). By contrast, a variant in which the tyrosine residues within the FUS IDR were substituted with serine[22] (FUSm-IDR–FREE1) lost the ability to phase separate in vitro (Extended Data Fig. 2j) and did not rescue *free1* lethality (Extended Data Fig. 3e). Taken together, these data indicate that IDR-mediated phase separation is essential for FREE1 function.

## Condensation enhances FREE1 binding to membranes

MVBs are membranous endosomal compartments, the small size of which is close to the optical resolution limit. Wortmannin treatment results in formation of functional MVBs of increased size, facilitating their visualization by light microscopy[24,25]. In wortmannin-treated seedlings, FREE1 formed condensates on MVBs (Fig. 2a and Extended Data Fig. 4a,b) that dissolved and reappeared after 1,6-hexanediol treatment and subsequent washout (Fig. 2b).

FREE1 has a FYVE domain (Fig. 1a) that is known to bind to the membrane lipid phosphatidylinositol 3-phosphate (PI3P) and to localize

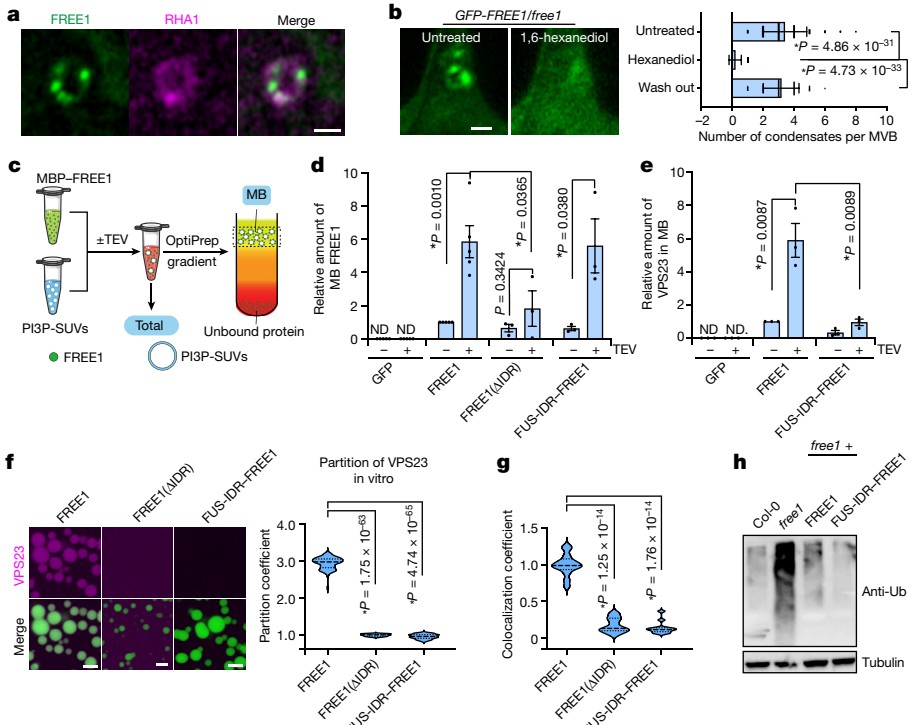

**Fig. 2 | FREE1 condensates bind to membranes and partition ESCRTs.**
**a**, FREE1 condensation on wortmannin-induced large MVBs in *A. thaliana* root tip cells expressing GFP–FREE1 and the MVB marker mCherry–RHA1. **b**, Treatment of wortmannin-enlarged MVBs as described in **a** with 1,6-hexanediol (left). Right, the number of FREE1 condensates per MVB. Data are mean ± s.d. *P* values were calculated using two-tailed *t*-tests; asterisks indicate significant differences. *n* = 72, 68 and 68 protoplasts, respectively. **c**, Overview of the membrane flotation assay. **d**, Quantification of membrane-bound (MB) proteins indicated in **c** as determined by immunoblotting. ND, no signal detected. Data are mean ± s.e.m. *P* values were calculated using two-tailed *t*-tests. *n* = 3 experiments. The gel blot is shown in Extended Data Fig. 4d. **e**, Enhanced VPS23 association with membranes in the presence of FREE1 but not FUS-IDR–FREE1 condensates. The assay was performed as in **d**. Data are mean ± s.e.m. *P* values

were calculated using two-tailed *t*-tests. *n* = 3 experiments. **f**, VPS23 (mCherry labelled) partitioning by FREE1 condensates in vitro. FREE1(ΔIDR) condensates formed at 10× concentration compared with FREE1. Data are mean ± s.d. *P* values were calculated using two-tailed *t*-tests; asterisks indicate significant differences. *n* = 45, 30 and 45 condensates, respectively. **g**, Colocalization of VPS23 (mCerulean labelled) with condensates of FREE1 variants in protoplasts. Data are mean ± s.d. *P* values were calculated using two-tailed *t*-tests; asterisks indicate significant differences. *n* = 14, 14 and 12 protoplasts, respectively. **h**, Accumulation of total ubiquitinated proteins in 10-day-old seedlings as determined by immunoblotting. Representative images of *n* = 3 (**a**) and *n* = 2 (**h**) independent experiments. Scale bars, 1 µm (**a** and **b**) and 5 µm (**f**). The unprocessed blots are provided in Supplementary Fig. 1. The schematic in **c** was created using Adobe Illustrator.

FREE1 to MVBs[6]. The FYVE domain is encoded outside the FREE1 IDR, and we found that FREE1 and FREE1(ΔIDR) displayed a similar binding affinity for PI3P over a range of concentrations (Extended Data Fig. 4c). To determine whether phase separation of FREE1 modulates its ability to bind to membranes, we used an in vitro flotation assay in which FREE1 was incubated with PI3P-containing small unilamellar vesicles (SUVs; Fig. 2c). While non-condensing variants (with a solubility tag, FREE1(ΔIDR)) bound similarly to the membrane fraction, condensing conditions (FREE1, FUS-IDR–FREE1) significantly enhanced FREE1 membrane association (Fig. 2d and Extended Data Fig. 4d). Consistent with the inhibitory effect of wortmannin on PI3P formation[26], we determined that cellular PI3P levels were reduced by a factor of five in our MVB enlargement experiments (Extended Data Fig. 4e,f). Importantly, reducing the PI3P concentration tenfold in our flotation assay still resulted in the enrichment of condensing FREE1 (Extended Data Fig. 4g). Furthermore, the expression of FREE1 variants in yeast, in which FYVE-containing proteins are known to localize to the vacuole membrane[27], resulted in high enrichment of FREE1 and FREE1(ΔIDR) on vacuole surfaces and in vacuole clustering (Extended Data Fig. 4h). Ablating the FYVE domain resulted in cytosolic condensate formation (FREE1(ΔFYVE)) or diffuse cytosolic localization (FREE1(ΔFYVEΔIDR)), confirming that the FYVE domain enhances membrane binding of FREE1 in vivo. Together, these data indicate that FREE1 condensates interact with membranes, which is consistent with a membrane-wetting interaction[11].

## FREE1 condensates partition ESCRTs

ESCRT-I subcomplex formation begins with associations between the VPS23, VPS37 and VPS28 proteins[28]. We found that these proteins are enriched by treatment with b-isox and contain IDRs, but do not phase separate in vitro (Extended Data Fig. 1). We hypothesized that FREE1 condensates function as scaffolds, whereas VPS proteins act as partitioning client molecules. Consistent with this, we observed that VPS23 association with SUVs largely increased when condensed FREE1 was present (Fig. 2e and Extended Data Fig. 5a), and that VPS23 partitioned in FREE1 condensates both in vitro and in protoplasts (Fig. 2f,g and Extended Data Fig. 5b). Consistent with this, VPS37 and VPS28, which act downstream of VPS23, were enriched significantly in VPS23-containing but not VPS23-free condensates (Extended Data Fig. 5c). A yeast two-hybrid assay (Extended Data Fig. 5d) confirmed a previous report showing that a motif within the FREE1 IDR interacts with VPS23[8]. In agreement with these data, VPS23 did not interact with FREE1(ΔIDR) and FUS-IDR–FREE1 condensates in partitioning assays (Fig. 2e–g and Extended Data Fig. 5a,b) and did not localize to MVBs in *FUS-IDR–FREE1/free1* protoplasts (Extended Data Fig. 5e). Together, these results indicate that proteins can be enriched sequentially by partitioning in FREE1 condensates, including components of the ESCRT machinery. Notably, we also found that expression of the non-VPS23 interacting variant FUS-IDR–FREE1 fully rescued the accumulation of ubiquitinated proteins in *FREE1*-knockout cells (Fig. 2h), suggesting

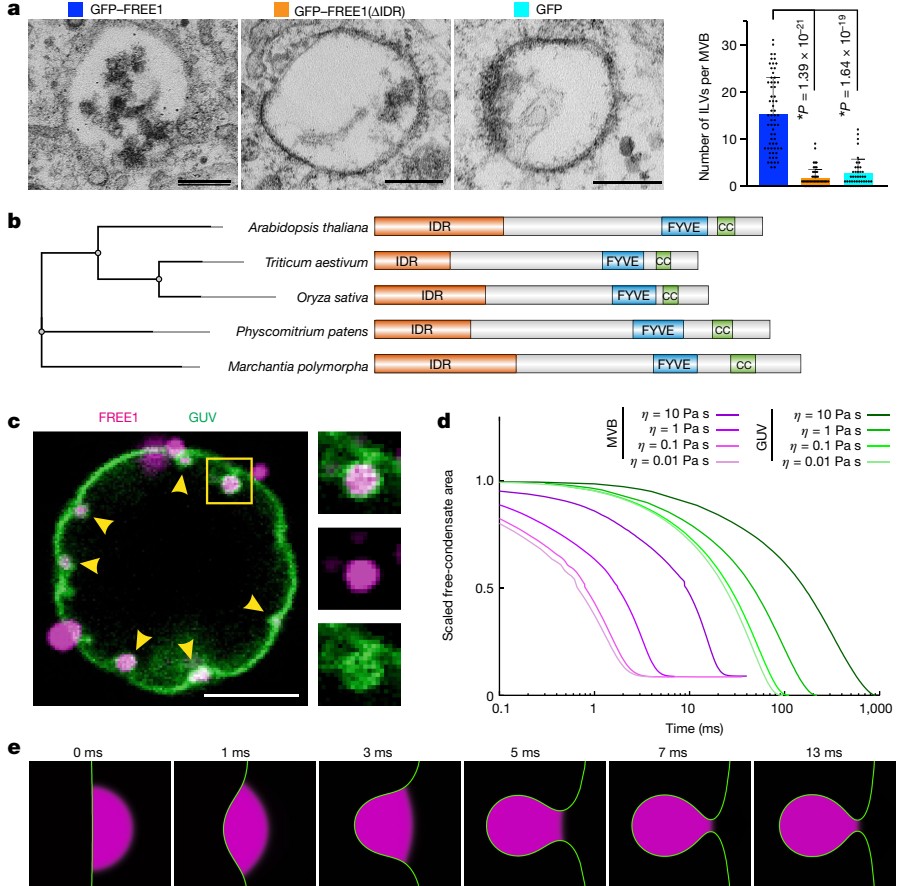

**Fig. 3 | FREE1 condensates mediate ILV biogenesis by membrane wetting.**
**a**, Mammalian COS-7 cells expressing the indicated proteins. Left, representative transmission EM (TEM) images. Scale bars, 100 nm. Right, ILV quantification. Data are mean ± s.d. $P$ values were calculated using two-tailed $t$-tests; asterisks indicate significant differences. $n = 64, 54$ and $41$ cells, respectively. **b**, Abbreviated phylogenetic tree of FREE1 homologues (Extended Data Fig. 6a) and the domain structure of FREE1 homologues. CC, coiled-coil domain. **c**, FREE1 condensates (magenta) mediate ILV-like invaginations (arrowheads) of GUV membranes (green). Scale bar, 5 μm. Inset: magnified ILV with membrane neck. $n = 3$ independent experiments. **d**, Dynamics of ILV formation in silico for varying ambient viscosities. The scaled free-condensate area (condensate/fluid surface area with respect to initial value) decreases over time. $\theta = 70°$; membrane tension $\sigma_{\alpha\gamma} = 1.5$ mN m$^{-1}$; membrane tension $\sigma_{\alpha\beta} =$ surface tension $\sigma_{\beta\gamma} = 1.1$ mN m$^{-1}$. Condensate viscosity $\eta_{cond} = 10$ Pa s; ambient (cytosol or buffer) viscosity $\eta_{ambient} = 0.01–10$ Pa s. Condensate diameters are 2 μm and 35 nm in the GUV-like and MVB-like simulations, respectively. **e**, MVB-ILV formation in silico time series for conditions as in **d** with $\eta_{cond}/\eta_{ambient} = 1$ and line tension $\lambda = 10$ pN. The neck radius is 1.5 nm.

that condensing FREE1 can function in cargo sorting and degradation, even in the absence of downstream ESCRTs.

## FREE1 is functional in several species

Early studies reported that FREE1 is restricted to eudicots, a subgroup of plant species[6]. Meanwhile, other ESCRT-0 proteins like HRS are conserved from budding yeast to humans; similarly to FREE1, these proteins have FYVE domains with the ability to bind to downstream ESCRTs and ubiquitin[29,30]. Although FREE1 has no direct homologue in mammals[6], it forms condensates after expression in mammalian cells (Extended Data Fig. 2f). In this context, we next tested whether FREE1 can contribute to mammalian MVB formation. Consistent with our plant data (Fig. 2a), immuno-gold electron microscopy (EM) analysis revealed FREE1 localization to mammalian MVB membranes (Extended Data Fig. 9a). Furthermore, we observed a significant increase in the number and size of ILV-like structures inside MVBs after expression of condensing FREE1 compared with the non-condensing controls (Fig. 3a). As these results indicate a common physiological function of FREE1, we performed a detailed phylogenetic analysis, identifying homologues in a broad range of land plants (Fig. 3b and Extended Data Fig. 6a,b). These FREE1 homologues are highly enriched by treatment with b-isox[17]

(Extended Data Fig. 6c) and combine conserved FYVE and coiled-coil domains with non-conserved N-terminal IDRs (Fig. 3b and Extended Data Fig. 6b,d). This suggests that the sequence divergence within these IDRs is unlikely to affect key protein functions, which is consistent with the ability of chimeric FUS-IDR–FREE1 and FLOE1-IDR–FREE1 to rescue *free1* seedling lethality (Fig. 1f and Extended Data Fig. 3c–e). In summary, these results indicate that FREE1 condensates can function in MVB biogenesis across species.

## Dynamics of membrane bending by condensates

As FREE1 condensates can bind to membranes and modulate MVB biogenesis in the absence of downstream ESCRTs, we speculated that FREE1 membrane wetting and capillarity may underlie ILV invagination[31]. Owing to their small size, ILVs cannot be resolved using optical live-cell imaging techniques (Fig. 3a). To understand how condensates can contribute to ILV formation, we developed a strategy that enabled us to assess ILV dynamics and shapes by coupling in vitro and in silico experimentation with mathematical modelling. We exposed PI3P-containing giant unilamellar vesicles (GUVs) to purified, phase-separating FREE1, which resulted in a near-immediate formation of numerous small, submicrometre-sized FREE1 condensates on GUVs.

These condensates induced local membrane invaginations and FREE1 condensate-filled vesicle-like structures that remained attached to the GUV through membrane necks (Fig. 3c). These condensate–GUV interactions are consistent with our observations of condensate-SUV flotation assays (Fig. 2d) and ILV induction after mammalian expression of FREE1 (Fig. 3a). Furthermore, archetypical condensate wetting morphologies[10,11] arising in these experiments suggest that such membrane invaginations may be analogous to the early stages of ILV formation.

To establish intermediate ILV morphologies and dynamics in more detail, we devised an in silico simulation based on the Navier–Stokes–Cahn–Hilliard equations. The underlying model considers the condensate surface to be a diffuse interface of surface tension $\sigma_{\beta\gamma}$, while the membrane is assumed to be an incompressible fluid characterized by bending stiffness $\kappa$ as well as membrane tension. Condensate-membrane adhesion (wettability) is described by the contact angle $\theta$ (Extended Data Fig. 7a and Supplementary Note 1). We experimentally determined that $\theta = 70°$ through observation of micrometre-sized wetting condensates that form lens-like shapes on GUVs (Extended Data Fig. 7b). Application of this empirically ascertained $\theta$ in silico confirmed that condensate–membrane contacts result in lens-like condensate morphologies for conditions with limited membrane excess area (Extended Data Fig. 7c). For conditions in which the membrane excess area equals the surface area of the condensate (Extended Data Fig. 7d), our simulations described the dynamics of membrane invagination, revealing that wetting interactions allow MVBs to form ILV-like structures with stable necks spontaneously within milliseconds, while generation of ILV-like structures in GUVs requires up to a second (Fig. 3d and Supplementary Videos 1 and 2). The differences in timescale reflect diverging sizes of interacting membranes and condensates. Furthermore, the timescale is influenced by the viscosity ratio discrepancy between membrane-contacting liquids (that is, the condensate and surrounding non-condensate solution; Extended Data Fig. 7e and Supplementary Video 3). Notably, both factors also influence ILV intermediate shapes (Extended Data Fig. 7f). Consistent with simple energy balance and dimensional analysis, we observed that condensates with diameter $D_{ILV}$ form ILV-like invaginations when the dimensionless parameter $C = \sigma_{\beta\gamma}D_{ILV}^2/\kappa$ is larger than 1 (Extended Data Fig. 7g,h). In summary, our computational model describes condensate-mediated membrane remodelling dynamics and indicates that membrane invaginations form as soon as the energetic favourability of condensate–membrane wetting overcomes the costs of membrane bending.

## Condensates mediate membrane scission

The ESCRT machinery is known to be essential for completing ILV biogenesis by membrane scission[1]. Scission can be understood as a consequence of membrane neck instability. Adhesion of particles to membranes has recently been reported to cause such instabilities[32] and spontaneous neck scission has been experimentally observed for a critical positive neck constriction force ($f^*$) above a value in the range of 25 pN (ref. 33). To understand the physical bases of condensate-mediated instability, we developed a mathematical model that examines the radius of the membrane neck ($R_{ne}$) and its effect on the energetic favourability of ILV scission. This energy $E(R_{ne})$ includes the contributions of wetting, membrane bending and surface-tension energies, as well as the line tension $\lambda$ of the three-phase contact line between the cytosol, FREE1 condensate and MVB membrane (Supplementary Note 2). Line tensions in the order of 1–100 pN are typical for three-phase contact lines[34,35], and can become the dominant force at the nanometre scale that characterizes membrane necks. The model predicts that a stable ILV neck can arise with positive line tension. This is corroborated by computer simulations, in which including $\lambda = 10$ pN reduces $R_{ne}$ from 6.5 nm to 1.5 nm (Fig. 3e and Supplementary Video 4) and increasing $\lambda$ results in notable acceleration

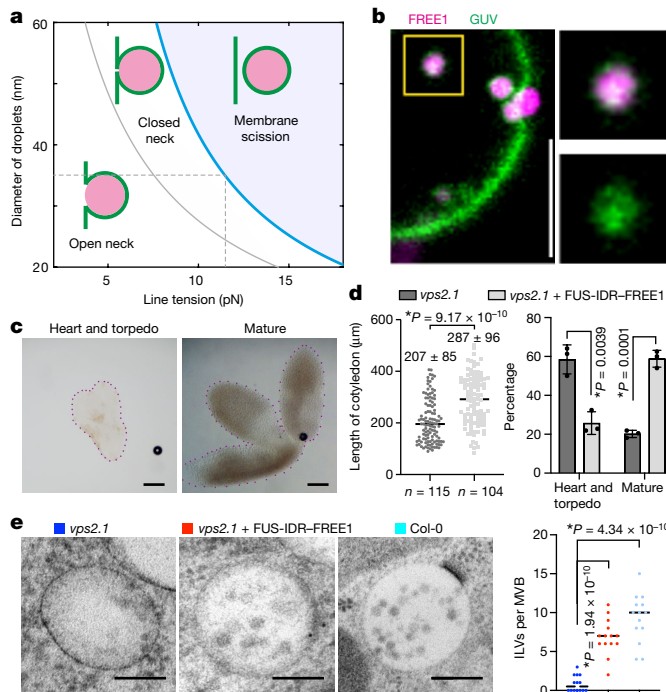

**Fig. 4 | FREE1 condensates scission membranes. a**, Theoretical stability diagram of condensate-induced membrane scission as a function of line tension and variable ILV size. The blue line indicates the critical scission line tension $\lambda^*$. Critical neck constriction force $f^* = 25$ pN; bending rigidity $\kappa = 10^{-19}$ J; $D_{MVB} = 200$ nm. The grey dashed line indicates $D_{ILV} = 35$ nm. **b**, FREE1-condensate-filled (magenta) ILV freely diffusing inside the GUV (green) shown in Fig. 3c at a later timepoint. Inset: magnified ILV. Confocal section. Scale bar, 5 μm. **c**, Representative embryos at the indicated stages. Scale bars, 50 μm. **d**, Statistical analyses of the length of cotyledons and percentage of embryos arrested at different stages. The black lines indicate the median (left). Data are mean ± s.d. (right). $P$ values were calculated using two-tailed $t$-tests; asterisks indicate significant differences. $n = 3$ experiments. **e**, Left, TEM images of MVBs in early-stage embryos of the indicated genotypes. Scale bars, 100 nm. Right, quantification of the number of ILVs per MVB. The black lines indicate the median. $P$ values were calculated using two-tailed $t$-tests; asterisks indicate significant differences. $n = 14$ MVBs. Representative images of $n = 3$ independent experiments (**b** and **c**).

of ILV formation (Extended Data Fig. 8a). Using $D_{ILV} = 35$ nm and $D_{MVB} = 200$ nm (Extended Data Fig. 9c) and $f^* = 25$ pN, our model predicts condensate-mediated scission occurring above a critical scission line tension of $\lambda^* = 11.3$ pN (Fig. 4a). Notably, smaller MVBs, larger condensates and negative membrane asymmetries reduce $\lambda^*$ (Fig. 4a and Extended Data Fig. 8b–f), and membrane-wetting condensates have been shown to induce such scission-promoting negative membrane asymmetries[36]. Consistent with the predictions of our model, we observed FREE1-condensate-filled ILV-like structures freely diffusing within GUVs in ESCRT-machinery-free GUV experiments (Fig. 4b). It has been shown that free diffusion of such vesicles indicates successful neck scission[37], suggesting that FREE1 condensates alone can mediate vesicle scission in vitro (Fig. 4b). Taken together, theoretical and experimental results suggest that the physical properties inherent to membrane wetting, line tension force and negative membrane asymmetry can scission membranes. Consistent with this, we observed that expression of condensing FREE1 in plants and mammalian cells resulted in condensate formation within MVBs (Extended Data Figs. 4b and 9b), while non-condensing FREE1(ΔIDR) was excluded from the MVB lumen (Extended Data Fig. 9b). Furthermore, *A. thaliana* root cells expressing condensing FREE1 produced larger ILVs in MVBs compared with cells expressing non-condensing FREE1 (Extended Data Fig. 9c).

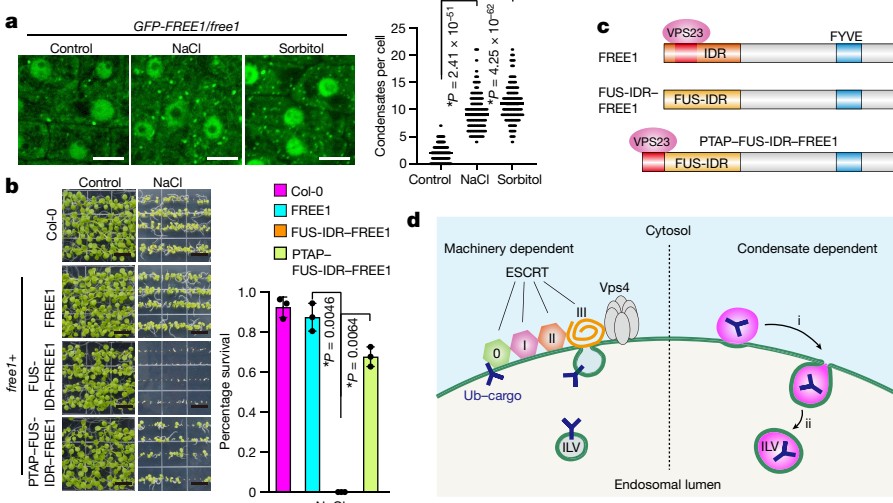

**Fig. 5 | FREE1 condensation is indispensable for osmotic tolerance.**
**a**, *A. thaliana* root tip cells expressing FREE1 were subjected to acute hyperosmotic treatments. Left, confocal sections. Right, the number of FREE1 condensates per cell. The black lines indicate the median. *P* values were calculated using two-tailed *t*-tests; asterisks indicate significant differences. *n* = 105 cells. **b**, *A. thaliana* seedlings germinating and growing under hyperosmotic conditions. Left, images of plants at 10 days. Right, survival rates. Data are mean ± s.d. *P* values were calculated using two-tailed *t*-tests; asterisks indicate significant differences. *n* = 3 experiments. Scale bars, 5 μm (**a**) and 1 cm (**b**). **c**, Diagram of FREE1-variant domain structures. The red sections indicate the PTAP-like motifs that mediate interactions with VPS23. **d**, A model of ESCRT-machinery-dependent and FREE1-condensate-mediated (magenta) ILV formation pathways from MVB membranes (green). i, membrane invagination; ii, membrane scission. Blue Y shapes indicate ubiquitinated cargoes. The schematic in **d** was created using Adobe Illustrator.

## FREE1 condensates rescue ESCRT defects

We next examined whether FREE1 condensates can rescue ILV formation in cells with a defective ESCRT machinery. VPS2 is a conserved subunit of ESCRT-III that is essential for ILV biogenesis[38]. In plants, germinating *vps2.1* homozygous knockouts arrest at early embryogenesis stages[39]. We found that around 60% of *A. thaliana vps2.1*-knockout embryos arrested at the early heart and torpedo stages (Fig. 4c,d), and that MVBs observed in these embryos contained very few ILVs (Fig. 4e). Importantly, overexpression of FUS-IDR–FREE1 in *vps2.1* knockouts (Extended Data Fig. 3f) largely rescued this ILV biogenesis defect (Fig. 4e). In these cells, we detected FUS-IDR–FREE1 inside MVBs by immuno-gold EM (Extended Data Fig. 9d), a result agreeing with previous reports of MVB-mediated FREE1 internalization and degradation[40]. Importantly, the size of FUS-IDR–FREE1-overexpressing knockout embryos increased by about 50%, with a significant increase in the number of embryos reaching a mature developmental stage (Fig. 4d). These findings indicate that FREE1 condensates contribute to MVB biogenesis by functionally compensating for defects in the ESCRT membrane scission machinery.

## FREE1 condensation enhances osmotolerance

MVB-mediated protein sorting and degradation have a key role in abiotic stress responses by ensuring the correct delivery and degradation of stress-related cargo molecules[41–43]. To understand the role of FREE1 in adverse environments, we examined FREE1 function under hyperosmotic conditions. We observed a significant increase in FREE1 condensate number (Fig. 5a) and FREE1-derived degradation products (Extended Data Fig. 10a,b), confirming FREE1 degradation under stress[40]. This is probably a physiological manifestation of the FREE1-mediated ILV formation mechanism described above.

Intracellular levels of the key plant stress hormone abscisic acid (ABA) increase under stress to promote the transcription of osmotic response-related genes[44]. Notably, we observed that *free1/FUS-IDR–FREE1* but not *free1/FREE1* plants were hypersensitive to exogenously added ABA (Extended Data Fig. 10c). Moreover, hyperosmotic stress reduced seed germination and seedling survival rates of *free1/FUS-IDR–FREE1* plants specifically to very low levels (Fig. 5b and Extended Data Fig. 10d). As the VPS23-binding PTAP-motif is present in the FREE1 IDR but lacking in the FUS IDR (Fig. 5c), we next tested whether an inability to interact with the ESCRT machinery through VPS23 causes the observed hypersensitivity of these plants to stress. After genetically introducing the PTAP motif to FUS-IDR–FREE1, we observed that PTAP–FUS-IDR–FREE1 interacted with VPS23 (Extended Data Fig. 10e,f), alleviated ABA sensitivity (Extended Data Fig. 10c) and rescued germination as well as seedling survival rates (Fig. 5b and Extended Data Fig. 10d). These results suggest that, while FREE1 condensation alone is sufficient for plant survival under standard conditions (Fig. 1f), osmotic tolerance requires a functional link between condensate-dependent and ESCRT-machinery-dependent MVB pathways (Fig. 5d).

Collectively, our findings demonstrate that FREE1 is a key phase-separating component of the MVB pathway that can act either alone or by recruiting binding partners, including soluble cargo proteins and ESCRT proteins. Partitioning of ESCRTs by FREE1 condensates probably facilitates ESCRT machinery assembly by increasing the local concentration of ESCRTs on MVB membranes. Alternatively, FREE1 condensates may sequester ESCRT proteins for degradation and recycling[40]. Our combined theoretical and experimental approaches reveal that FREE1 condensate-mediated ILV formation can proceed in two steps: capillary forces initially drive MVB membrane invagination, following which wetting-related line tension forces mediate ILV scission (Fig. 5d). Mechanistically, FREE1 condensates constrict membrane necks indirectly from within the forming vesicle, contrasting the direct neck constriction by neck-localizing filamentous ESCRT machinery spirals. Consistent with the proposed scission mechanism, localization of FREE1 as cargo within autophagosomes has been shown to promote autophagosomal membrane scission[45].

Biomolecular condensates are increasingly recognised as central players in a range of trafficking processes that depend on membrane scission. Examples include not only ILV formation, as discussed here, but also endocytic machinery condensates on yeast and plant plasma membranes that can support endocytosis[46,47], cytosolic condensates that mediate their own internalization and degradation through

lysosome-related pathways[12,48,49], vacuolar storage protein condensates that form during plant embryogenesis before tonoplast scission occurs[13] and condensate-containing secretory granules that scission from within the trans-Golgi network[50]. This study provides multiple lines of evidence demonstrating that wetting condensates can mediate the membrane shaping and scission processes possibly underlying these and other trafficking processes. Shaping dynamics and physical parameters provided by our in silico simulations and theoretical modelling will allow for a detailed understanding of diverse condensation-mediated membrane morphologies (Extended Data Fig. 8d–f). Furthermore, the scission mechanism proposed here could have implications for our understanding of the evolution of compartmentalization and the potential interplay between conventional machinery-dependent and condensate-mediated cellular scission pathways[1,16,51] (Fig. 5d). Condensates also could provide a means to hinder free diffusion inside ILVs[52]. Together, this study identifies that, beyond their established function in capillary phenomena[10,11], condensates can mediate membrane scission events and have a key role in cellular trafficking processes.

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

# Methods

## Plant materials and growth conditions

*A. thaliana free1* mutant (a T-DNA insertion line) seeds and *UBQ10::GFP–FREE1* transgenic seeds were provided by L. Jiang. All other transgenic *A. thaliana* lines were generated by transformation of corresponding constructs into the heterozygous *free1* mutant background and subsequent genotyping of homozygous *free1* from progeny. The *ups2* mutant (a T-DNA insertion line) seeds were provided by Y. Cheng. The RHA1–mCherry marker line was gifted by L. Qu. Seeds were surface sterilized by ethanol and sown on half-strength Murashige and Skoog (MS) medium with 1.0% sucrose, 0.4% phytagel at pH 5.8. MS was supplemented with NaCl or sorbitol as indicated. Plate media were stratified at 4 °C for 2 days and transferred to a growth chamber under a long-day (16 h–8 h light (22 °C)–dark (18 °C)) photoperiod.

## Chemical and osmotic seedling treatments

Wortmannin-treated seedlings were prepared by soaking for 45 min in liquid MS medium with wortmannin (Selleck, S2758) added at a final concentration of 33 μM from a 33 mM stock solution in dimethyl sulfoxide. For hexanediol treatment, 1,6-hexanediol was added at a final concentration of 4% (w/v) to MS medium with wortmannin (as above) and incubated for 5 min before imaging. For washout experiments, seedlings were incubated in half-strength MS medium with wortmannin (as above) without 1,6-hexanediol for 1 h before imaging.

For acute NaCl and sorbitol treatments, seedlings were soaked in liquid MS medium supplemented with 125 mM NaCl or 300 mM sorbitol for the indicated times before imaging. To measure germination and seedling survival rates, seeds were sown on MS plate medium supplemented with 125 mM NaCl or 300 mM sorbitol and grown at 22 °C under a long-day (16 h–8 h light (22 °C)–dark (18 °C)) photoperiod for 10 days.

## Plasmid construction

To generate the constructs for in vitro protein expression, the coding sequences of FREE1 and FREE1 variants (FREE1(ΔIDR), FUS-IDR–FREE1, PTAP–FUS-IDR–FREE1, FUSm-IDR–FREE1 and FLOE1-IDR–FREE1) were amplified and inserted into the pET11-6×His-MBP[53] or pET11-6×His vector. The coding sequences of VPS23A, VPS28A, VPS37A, TOL6, TOL9, LIP5 and BRO1 were amplified and inserted into the pRSFduet-6×His-mCherry vector. All cloning was performed using the ClonExpress II One Step Cloning kit (Vazyme, C112).

To generate constructs for transient expression in *A. thaliana* protoplasts, the coding sequences of FREE1 and FREE1 variants (FREE1(ΔIDR), FUS-IDR–FREE1), and RHA1 were amplified and inserted into modified pBI221 vectors containing a GFP tag or an mCherry tag, respectively.

For complementation constructs, an approximately 1 kb *FREE1* promoter region was amplified and inserted into the pCambia1300-N1-GFP vector to generate the pCambia1300-pFREE1-GFP construct. Moreover, the coding sequences of FREE1 and FREE1 variants (FREE1(ΔIDR), FUS-IDR–FREE1, PTAP–FUS-IDR–FREE1, FUSm-IDR–FREE1 and FLOE1-IDR–FREE1) were amplified and inserted into pCambia1300-pFREE1-GFP. All of the resulting constructs were introduced into *Agrobacterium tumefaciens* strain GV3101 and transformed into the *free1* heterozygous mutant using the floral dip method. In brief, *A. tumefaciens* GV3101 cells were collected and resuspended in a 5% sucrose solution containing 0.02% (v/v) silwet L-77. *A. thaliana* flower buds were dipped into the *A. tumefaciens* suspension for 2 min and subsequently kept in the dark overnight before being transferred to the growth chamber.

For the b-isox precipitation assay, proteins were transiently expressed in *Nicotiana benthamiana*. The coding sequences of VPS23A, VPS28A, VPS37A, TOL6, TOL9, LIP5 and BRO1 were amplified and inserted into pCambia1300-N1-Flag vector. All of the constructs were introduced into the *A. tumefaciens* strain GV3101 and infiltrated into *N. benthamiana* leaves.

For yeast two-hybrid constructs, the coding sequences of FREE1 variants and VPS23A were amplified and inserted between the EcoRI and BamHI restriction sites of pGADT7 and pGBKT7 using the ClonExpress II One Step Cloning kit (Vazyme, C112). A list of all of the coding and primer sequences is provided in Supplementary Table 1.

## In vitro protein expression and purification

All fusion proteins were expressed and purified from *Escherichia coli* (Rosetta) cell extracts on a Ni-NTA column. In brief, protein expression was induced by 0.4 mM isopropyl-β-D-1-thiogalactopyranoside at 18 °C overnight. Cells were collected by centrifugation and resuspended in lysis buffer (40 mM Tris-HCl pH 7.4, 500 mM NaCl, 10% glycerol). The suspension was sonicated for 30 min (2 s on, 4 s off, SCIENTZ) and centrifuged at 13,000g for 30 min at 4 °C. The supernatant was incubated with Ni-NTA agarose for 20 min, washed and eluted with 40 mM Tris-HCl pH 7.4, 500 mM NaCl and 500 mM imidazole (Sangon). Proteins were further purified by gel-filtration chromatography (Superdex-200; GE Healthcare) and stored in 40 mM Tris-HCl pH 7.4, 50 or 500 mM NaCl, 1 mM dithiothreitol on ice or at −80 °C.

## In vitro phase-separation assay

The solubility tag MBP was cleaved using TEV protease for 30 min to induce LLPS. Protein concentrations were determined by measuring absorbance at 280 nm using a NanoDrop ND-1000 spectrophotometer (Thermo Fisher Scientific). Proteins were diluted to various salt (50–500 mM NaCl) and protein (0.05–20 μM) concentrations in 384-well plates (Greiner Bio One, 781090) and observed using the Zeiss LSM880 confocal laser-scanning microscope equipped with a ×63 objective.

To quantify the partition coefficient of VPS proteins by FREE1 condensates, regions of interest of equivalent size were analysed in the Fiji implementation of ImageJ to calculate the VPS signal intensity inside and outside FREE1 condensates. The partition coefficient was defined as intracondensate fluorescence intensity divided by the fluorescence intensity of the extracondensate solution.

## FRAP analysis

FRAP analysis was performed on the Olympus Fluoview FV-1000 confocal laser-scanning microscope. Condensates were photobleached using a 488 nm Ar-laser pulse at maximum intensity. Time-lapse recordings of intensity changes were analysed using the Fiji implementation of ImageJ.

## Transient expression in *A. thaliana* protoplasts

*A. thaliana* mesophyll protoplasts were isolated from 3-week-old Col-0 plants as previously described[54]. In brief, leaf slices from the middle part of a leaf were incubated in enzyme solution (20 mM MES pH 5.7, 1.5% (w/v) cellulase R10, 0.4% (w/v) macerozyme R10, 0.4 M mannitol and 20 mM KCl) for 3 h. The suspension was centrifuged and washed twice with W5 solution (2 mM MES pH 5.7, 154 mM NaCl, 125 mM CaCl$_2$ and 5 mM KCl). Plasmids were transformed into protoplasts using the PEG-calcium-mediated method (40% PEG3350). The transfected protoplasts were incubated at 22 °C for 16 h in WI solution (0.5 M mannitol, 4 mM MES pH 5.7 and 20 mM KCl) and observed under the Zeiss LSM880 confocal laser-scanning microscope equipped with a ×63 objective. The colocalization coefficient was defined as the ratio of VPS23 fluorescence intensity relative to FREE1 fluorescence intensity in the same condensate. Data analysis was performed in the Fiji implementation of ImageJ.

## b-isox precipitation

The precipitation of ESCRT proteins by b-isox was performed as described previously. In brief, 0.5 g of fine powder ground from tobacco leaves was lysed in extraction buffer (10 mM Tris-Cl pH 7.5, 150 mM NaCl, 5 mM MgCl$_2$, 20 mM β-mercaptoethanol, 0.5% NP-40;

10% glycerol, 1× protease inhibitor cocktail). The cell lysate was then clarified by centrifugation. Then, 500 μl protein extract was incubated with 30 μM b-isox at 4 °C for 1 h and centrifuged. The precipitate was washed twice with ice-cold extraction buffer and boiled in SDS loading buffer (100 mM Tris-HCl pH 7.4, 4% SDS, 0.2% bromophenol blue, 20% glycerol, 5% β-mercaptoethanol). For precipitation of FREE1, ten-day-old *A. thaliana* Col-0 seedlings were used. For precipitation of other ESCRT proteins, each Flag-tagged protein was transiently expressed and resulting tobacco cells were used for precipitation.

## Immunoblot analysis

To prepare total protein extracts, *A. thaliana* seedlings were ground in liquid nitrogen and lysed with extraction buffer (40 mM HEPES-KOH at pH 7.5, 1 mM EDTA, 10 mM KCl, 0.4 M sucrose with 1 × cOmplete Protease Inhibitor Cocktail). Debris was removed by centrifugation at 12,000*g* for 30 min. Protein samples were separated by 10% SDS−PAGE and transferred to a polyvinylidene difluoride membrane. Primary antibodies were used against FREE1, Flag (Merck, F1804, 1:2,000), GFP (Roche, 11814460001, 1:7,000), tubulin (Sigma-Aldrich, T5168, 1:2,000), ubiquitin (Santa Cruz Biotechnology, sc-8017; 1:1,000) or His-tag (Sangon, D110002, 1:1,000). The horseradish peroxidase (HRP)-conjugated secondary antibodies goat anti-Mouse (CWBIO, CW0102, dilute at 1:10,000) and goat anti-rabbit (CWBIO, CW0103, 1:10,000) were used for protein detection by chemiluminescence (ChemiDoc, LAS4000).

## Confocal microscopy

Confocal imaging was performed using the Zeiss LSM880 or Olympus Fluoview FV-1000 microscope equipped with ×63/1.4 NA oil and ×60/1.2 NA water-immersion objectives. GFP was excited at 488 nm and detected between 490 nm and 530 nm or between 500 nm and 544 nm. mCherry and DiIC18 were excited at 561 nm and detected at 579–650 nm or 570–670 nm.

## Cell culture and transfection

Maintenance of cell lines and transfection were performed as described previously[55]. COS-7 and HEK293T cells were maintained in DMEM supplemented with 10% FBS at 37 °C in 5% $CO_2$. Transfection was performed using X-tremeGENE HP (Roche, 28088300) according to the manufacturer's protocols.

## Lentiviral transduction

For lentiviral transduction, HEK293T cells were transfected with pFUGW (together with VSVG and psPAX2 plasmids). Viruses were harvested at 60–72 h post transfection, and the viral supernatant was centrifuged at 600 *g* for 5 min to remove cell debris. The indicated cells were infected with the viral supernatant diluted with fresh medium (30% viral supernatant) containing 10 μg ml$^{-1}$ polybrene.

## Immunofluorescence microscopy and quantification

After being infected with indicated lentiviruses for 72 h, the cells were fixed with 4% paraformaldehyde for 15 min at room temperature and used for imaging. For super-resolution microscopy, imaging experiments were performed using the Nikon combined confocal A1/SIM/STORM system with four excitation/imaging lasers (405, 488 and 561 nm from Coherent, 647 nm from MPBC) and a CFI Apo SR TIRF ×100/1.49 NA oil-immersion objective. Images were acquired using an Andor EMCCD camera (iXON 897). Data analyses were performed using the NIS-Elements AR (Nikon) software. All SIM images were acquired as *z*-stack images.

## TEM and immuno-gold labelling in mammalian cells

COS-7 cells were fixed with 2.5% glutaraldehyde for 1 h at room temperature and washed three times (15 min each) with 0.1 M PB. Post-fixation staining was performed with 1% osmium tetroxide (SPI, 1250423) for 30 min on ice. Cells were washed three times (for 15 min each) with

ultrapure water and placed in 1% aqueous uranyl acetate (EMS, 22400) at 4 °C overnight. The samples were then washed three times (15 min each) with ultrapure water, dehydrated in a cold-graded ethanol series (50%, 70%, 80%, 90%, 100%, 100%, 100%; 2 min each), and infiltrated with EPON 812 resin using 1:1 (v/v) resin and ethanol for 8 h, 2:1 (v/v) resin and ethanol for 8 h, 3:1 (v/v) resin and ethanol for 8 h, pure resin 2 × 8 h. After a final infiltration with fresh resin, the samples were polymerized at 60 °C for 48 h. Embedded samples were sliced into 75-nm-thick sections and stained with uranyl acetate and lead citrate (C1813156) before imaging on the HT-7800 120 kV transmission electron microscope (Hitachi High-Technologies).

Immuno-gold labelling was performed as described previously[56] with some modifications. In brief, COS-7 cells were incubated in 2% paraformaldehyde and 0.01% glutaraldehyde in PB buffer at 4 °C overnight and then washed with chilled PB/glycine. Cells were next scraped from the bottom of plastic dishes into 1% gelatine, centrifuged at 1,000 rpm for 2 min and resuspended in 12% gelatine at 37 °C for 10 min. The gelatine–cell mixture was then solidified on ice for 15 min. Small blocks (about 0.5 mm³) were cut and immersed in 2.3 M sucrose overnight at 4 °C. Then, 70-nm-thick cryosections were prepared at −120 °C with an ultramicrotome (Leica, EM FC7). After the sections were thawed at room temperature, immunolabelling was performed using rabbit anti-GFP antibodies followed by the immune-gold secondary antibody. The sections were then treated with methyl cellulose/uranyl acetate and subsequently imaged using the HT-7800 120 kV transmission electron microscope (Hitachi High-Technologies).

## Expression of FREE1 in *Saccharomyces cerevisiae*

*FREE1* ORFs were amplified and inserted into a pRS416 yeast expression vector containing the *GAP1* promotor sequence (pAM199). Vectors were transformed into SEY6210 wild-type cells (yAM007) and grown in SDCA medium (0.17% yeast nitrogen base without amino acids and ammonium sulphate, 0.5% ammonium sulphate, 0.5% casamino acids, 2% glucose, 100 μM L-histidine, 100 μM L-tryptophan) overnight. Cells were then diluted into fresh SDCA medium and grown for 4 h. Before imaging, vacuolar membranes were stained for 15 min in medium containing 160 μM FM4-64 dye, washed twice and chased for a further 30 min in fresh medium. Imaging was performed on the SpinSR10 spinning-disc confocal microscope, and the recorded images were deconvoluted using the Cellsens software (Olympus).

## Dot blot for PI3P-binding assay

PI3P lipids (Avanti Polar Lipids, 850187P) were spotted onto nitrocellulose membranes (Millipore) at the indicated concentrations (2%, 200 pmol; 1%, 100 pmol; 0.5%, 50 pmol; 0.1%, 10 pmol; 0.05%, 5 pmol). These membranes were then blocked in 5–10 ml blocking buffer (3% BSA in PBS-T buffer) for 1 h at room temperature, incubated with 10 nM of the indicated GFP-tagged protein (in blocking buffer) for 1 h at room temperature and washed three times (5 min each) with PBS-T buffer. GFP fluorescence was detected using the ChemiDoc imaging system (Bio-Rad).

## Membrane flotation assay

SUVs were prepared as described previously[55]. In brief, lipids were mixed as POPC:POPS:cholesterol:PtdIns(3)P:PE at a molar ratio of 62:10:25:1:1 in trichloromethane. The lipids were dried under a nitrogen stream and further dried for 1 h at 37 °C. The lipid film was then hydrated using Tris buffer (40 mM Tris HCl, pH 7.4, 150 mM NaCl) and subjected to 10 cycles of freezing in liquid nitrogen and thawing in a 42 °C water bath. Liposomes were extruded through a 400 nm pore size polycarbonate film to produce the SUVs. Proteins (100 nM) were then added to the SUV solution and incubated for 5 min at room temperature. To remove unbound protein, a membrane flotation procedure was then performed. For each 120 μl SUV–protein solution, 480 μl 50% OptiPrep was added. The mixture was overlaid with 480 μl 30% OptiPrep

and 90 μl Tris buffer (40 mM Tris HCl, pH 7.4, 50 mM NaCl). After centrifugation at 100,000g for 2 h, 60 μl of the top fraction containing the protein-bound SUVs was collected. To ensure equal loading of SUVs, 1 μl of the top fraction was used to measure the PC concentration using the Phospholipid C Kit (Wako, 433-36201). The top fraction as well as the total fraction were boiled in SDS loading buffer and used for immunoblot analyses.

### Yeast two-hybrid assay
The yeast two-hybrid assay was performed using the Matchmaker Gold Y2H system according to the manufacturer's instructions (Clontech). In brief, the constructs were co-transformed pairwise into yeast strain AH109 and cultured on SD/−Trp−Leu medium for 3 days. The interaction was analysed by spreading transformed cells ($10^3$−$10^7$ cells per ml) onto selective SD/−Trp−Leu−His−Ade medium supplemented with 5 mM 3-amino-1,2,4-triazole (3-AT).

### TEM analysis of plant samples
Observation of ILVs by TEM was performed as previously described. In brief, root tips from five-day-old *A. thaliana* seedlings or germinated seeds were subjected to high-pressure freezing (EM PACT2, Leica) and substituted by acetone containing 0.4% uranyl acetate at −85 °C overnight in an AFS freeze-substitution unit (Leica). Next, the samples were infiltrated with increasing concentrations of HM20 (33−66−100%), embedded and ultraviolet polymerized for 48−72 h. Ultrathin sections were cut on the Leica UC7 ultramicrotome. Ultrathin section on grids were observed using the 80 kV Hitachi H-7650 transmission electron microscope (Hitachi High-Technologies) equipped with a charge-coupled device camera.

### GUV preparation
GUVs were generated by the electroformation method[57] with modifications[58]. The lipid mixture contained 1-palmitoyl-2-oleoyl-glycero-3-phosphocholine (POPC, Avanti Polar Lipids) and 1,2-dioleoyl-*sn*-glycero-3-phospho-(1′-myo-inositol-3′-phosphate) (PI(3)P, Avanti Polar Lipids,) at a 9:1 molar ratio. Vesicles were fluorescently labelled by addition of 0.4 mol% membrane dye (DiIC18, Sigma-Aldrich). Lipid films were deposited onto indium-tin-oxide-coated glass plates (PGO) using glass syringes and 4 mM lipid stocks in chloroform, maintained for 1 h under vacuum and subsequently assembled into a chamber with a 2 mm Teflon spacer. The chamber was held together by binder clips, filled with 170 mM sucrose solution (Osmomat 3000 basic, Gonotec) connected to a function generator and an alternating current of 3.5 V and 10 Hz frequency was applied for approximately 2 h at 30 °C. GUVs were carefully collected, concentrated by sedimentation following a 1:10 dilution in isosmotic glucose solution and stored at room temperature until use.

### GUV wetting and ILV formation assays
FREE1 condensates were generated by TEV cleavage of MBP from FREE1-MBP in 96-well high-content imaging plates coated by applying and drying a 1% polyvinyl alcohol (146−186 kDa, Sigma-Aldrich) solution in water before use. To quantify FREE1 condensate−GUV contact angles, GUVs were added 1−2 min after micrometre-sized FREE1 condensates had formed in an iso-osmotic solution (40 mM Tris-HCl pH 8.0, 50 mM NaCl, 1 mM dithiothreitol). Confocal sections of condensate-GUV contacts were used to measure all three angles at the three-phase contact line and to compute the Young's contact angle (Extended Data Fig. 7b). For membrane fission assays, GUVs and the isosmotic TEV solution were first mixed before addition of FREE1−MBP and immediate imaging by confocal microscopy (Olympus Fluoview FV-1000) as described above.

### Computer simulations
The computational model incorporates three fluid phases: a fluid inside the vesicle and two immiscible fluids surrounding the vesicle.

The vesicle was modelled as a fluidic inextensible membrane with bending stiffness. The model includes the fluid motion of each of the three fluid phases coupled to the motion of the membrane. Membrane mechanics are governed by out of plane bending stiffness $\kappa$ and in-plane stretching elasticity. To solve the coupled physical system, we developed a numerical model based on the combination of the three-phase fluid model for wetting[59] and the elastic shell model[60] (further details are provided in Supplementary Note 1).

The stretching resistance of the membrane was assumed to be large enough that the membrane stretches locally less than 3%. An initially elongated membrane (Extended Data Fig. 7d) was put in contact with a half circular condensate with surface tension $\sigma_{\beta\gamma}$ (Supplementary Note 1), where the excess membrane area was set to allow for the condensate to be fully enclosed. Owing to the capillary forces of the condensate, both the membrane and droplet mutually remodel. The simulation model captures the full interaction of membrane bending stiffness, fluid motions and wetting deformations, enabling it to characterize intermediate shapes and time scales of the process.

### Theory of condensate-mediated membrane scission
To understand the conditions under which membrane necks close, the stability of such necks and the timing of their scission, the energy of the system $E(R_{ne})$, which depends on the radius of the neck $R_{ne}$, was considered. This energy includes contributions corresponding to the bending energy of the membrane (of which the properties are affected by the presence of the condensate), the wetting and surface tension energies of the condensate and the line tension of the three-phase contact line (Supplementary Note 2). The force on the closed neck was calculated as $f = \frac{dE(R_{ne})}{dR_{ne}}\Big|_{R_{ne}=0}$. For $f < 0$, the force acts to expand the neck, giving rise to either lens-like morphologies or endocytosis-like morphologies with wide necks (Fig. 4a). For $0 \le f \le f^*$, the force constricts the neck into a closed form, but is not large enough to induce scission. Finally, for $f > f^* \approx 25$ pN, strong constriction causes scission of the neck.

### Statistics and reproducibility
Sample sizes are chosen as widely used in the field. Biological and technical replicates were performed as described in the Methods for each experiment and conform to standards in the field. Exact *n* numbers for each experiment are provided in each figure legend. No data were excluded from analysis. Randomization of samples was performed. Seedlings from different plates were collected. Blinding was not deemed necessary in our study because we made no a priori assumptions on the response of the different samples to the experimental treatment, samples were all treated in parallel and all samples treated were always measured. Cells used in this study have been tested negative for mycoplasma contamination.

### Reporting summary
Further information on research design is available in the Nature Portfolio Reporting Summary linked to this article.

## Data availability
All data are available in the Article and its Supplementary Information. The full versions of all gels and blots are provided in Supplementary Fig. 1. Figures describing the results of the model can be reconstructed directly from the equations and procedures presented in the paper and in the Supplementary Methods, and require no data. Source data are provided with this paper.

## Code availability
The numerical simulation code is available at Zenodo[61] (https://doi.org/10.5281/zenodo.11919014). Installation requires the finite

element library AMDiS[62], which is available online (https://gitlab.math.tu-dresden.de/iwr/amdis).

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

**Acknowledgements** We thank Y. Cao and J. Chai (Westlake University) for the preparation of recombinant proteins; L. Qu (Peking University) for sharing the RHA1 marker line; Y. Cheng (Chinese Academy of Sciences) for sharing the *vps2* T-DNA line mutant; and L. Ge (Tsinghua University) for providing technical assistance. This work was supported by grants from the Beijing Natural Science Foundation (JQ21020) and the National Natural Science Foundation of China (32161133001, 32222015) to X.F.; Deutsche Forschungsgemeinschaft grants 460056461 to R.L.K. and 506366351 to R.L.K. and J.A.-C.; grants AL1705/3 and AL1705/5 to S.A.; Research Grants Council of Hong Kong (C4002-20W, C4014-23G, SRF2122-4S01 and CRS_CUHK405/23) to L.J.; and JSPS KAKENHI grant JP21K15083 to A.I.M. This work was supported in part by the World Research Hub (WRH) Program of the International Research Frontiers Initiative, Tokyo Institute of Technology.

**Author contributions** X.F. and R.L.K. conceived, guided and supervised the project. Y.W. designed and performed all in planta experiments and in vitro phase-separation assays. S.L. performed membrane flotation assays and mammalian expression experiments. S.A. and M.M. developed and performed computer simulations. J.A.-C. developed the theoretical model. Y.Z., Z.L., W.W. and L.J. performed the TEM experiments. H.Z. performed the yeast two-hybrid assay. A.I.M. performed yeast FREE1 expression experiments. F.Y. provided the antibodies against ubiquitin. R.L.K. and K.S. performed GUV assays. X.F., R.L.K. and A.I.M. wrote the manuscript. All of the authors discussed the results and contributed to the writing and review of the manuscript.

**Funding** Open access funding provided by Universität zu Köln.

**Competing interests** The authors declare no competing interests.

**Additional information**
**Correspondence and requests for materials** should be addressed to Roland L. Knorr or Xiaofeng Fang.

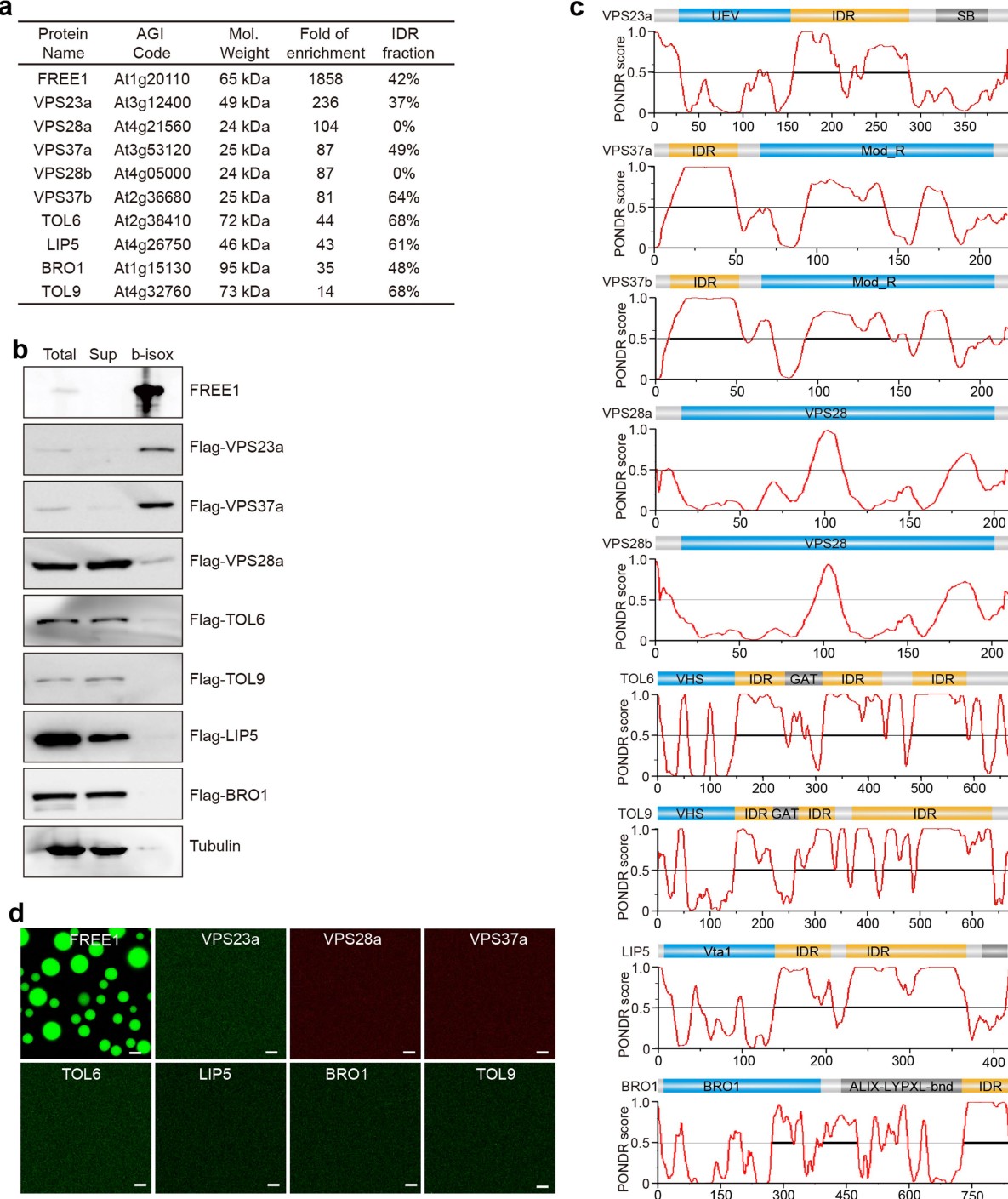

**Extended Data Fig. 1 | Identification of FREE1 as a phase separation-prone protein. a**, ESCRT components enriched by b-isox treatment of ten-day-old *A. thaliana* Col-0 seedlings (FREE1) or *N. bethamiana* leaves transiently expressing FLAG-tagged proteins. **b**, Immunoblot analyses of ESCRT proteins in total cell extracts, the lysates following b-isox enrichment (Sup) and b-isox precipitates using FREE1 or FLAG antibodies. Tubulin served as a loading control. **c**, Predicted intrinsically disordered regions by PONDR. **d**, Confocal microscopy images of GFP–FREE1 and mCherry-tagged ESCRT proteins at 5 µM concentration in 40 mM Tris-HCl pH 7.4, 150 mM NaCl. Scale bars, 5 µm. Representative images of 3 independent experiments. The unprocessed blots are provided in SI Fig. 1.

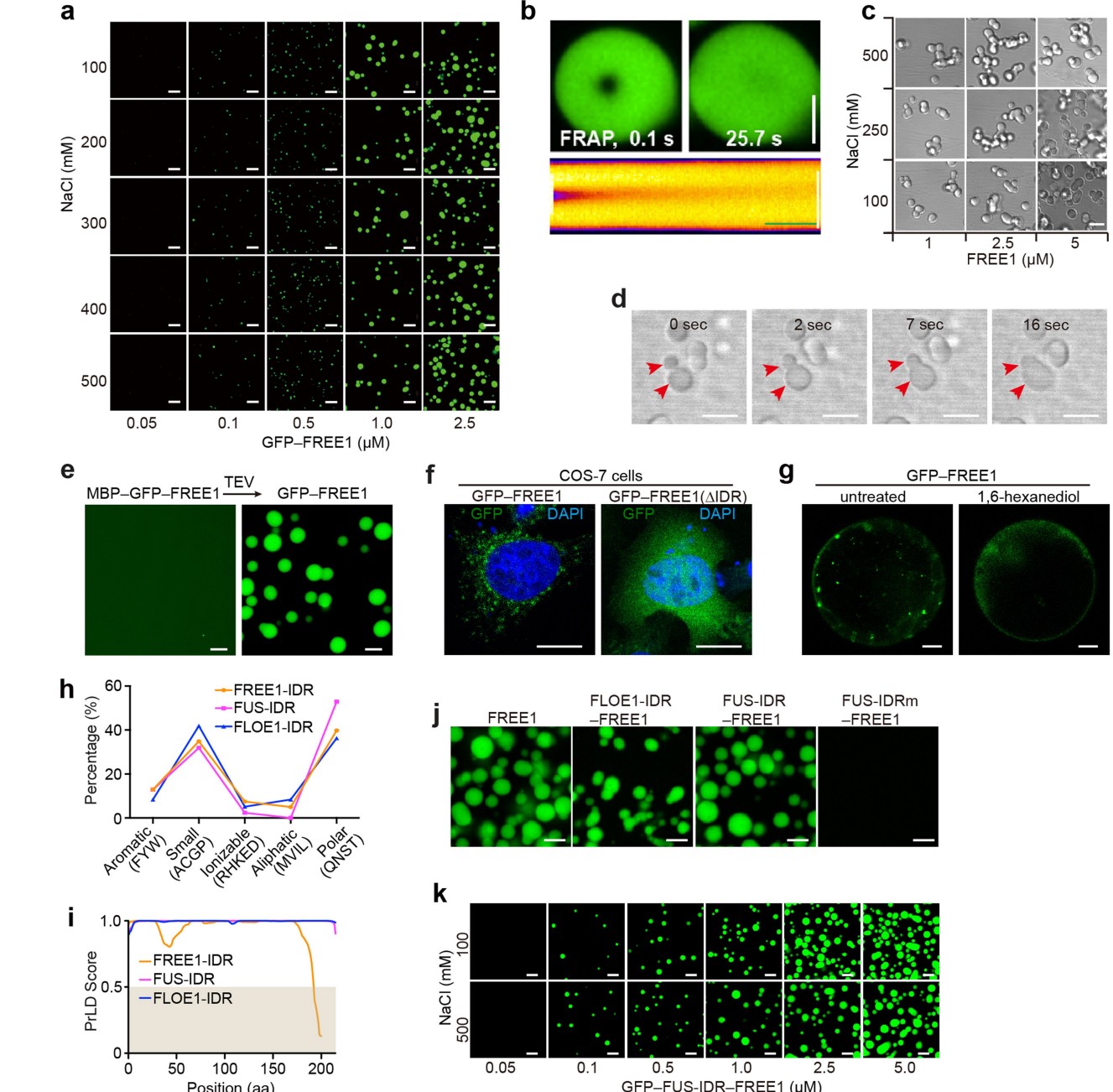

**Extended Data Fig. 2 | FREE1 and its variants phase separate in vitro and in vivo. a,b,k,** Confocal microscopy images of GFP–FREE1 (**a**), unlabelled FREE1 (**b**), and GFP–FUS-IDR–FREE1 (**k**) condensates formed at indicated protein and salt concentrations. Scale bars, 5 µm. **c,** Time lapse imaging of unlabelled FREE1 condensate coalescence. Scale bars, 5 µm. **d,** FRAP of FREE1 condensates. Top, first and last post-FRAP frames. Bottom, recovery kymograph (fire lookup table). Green scale bar, 5 s. **e,** Removal of MBP by TEV cleavage allows FREE1 LLPS.

**f,** Treatment of FREE1 condensates in *A. thaliana* protoplast cells with 1,6-hexanediol. **g,** FREE1 condensates in COS-7 cells. **h,** Percentage of different amino acids classes within IDRs used in this study. **i,** Disorder of the prion-like domain predicted by the PLAAC algorithm. **j,** In vitro phase separation assay of 10 µM GFP-labelled FREE1 variants. Representative images of *n* independent experiments (*n* = 3 (**a-g,j,k**)). Scale bars, 5 µm.

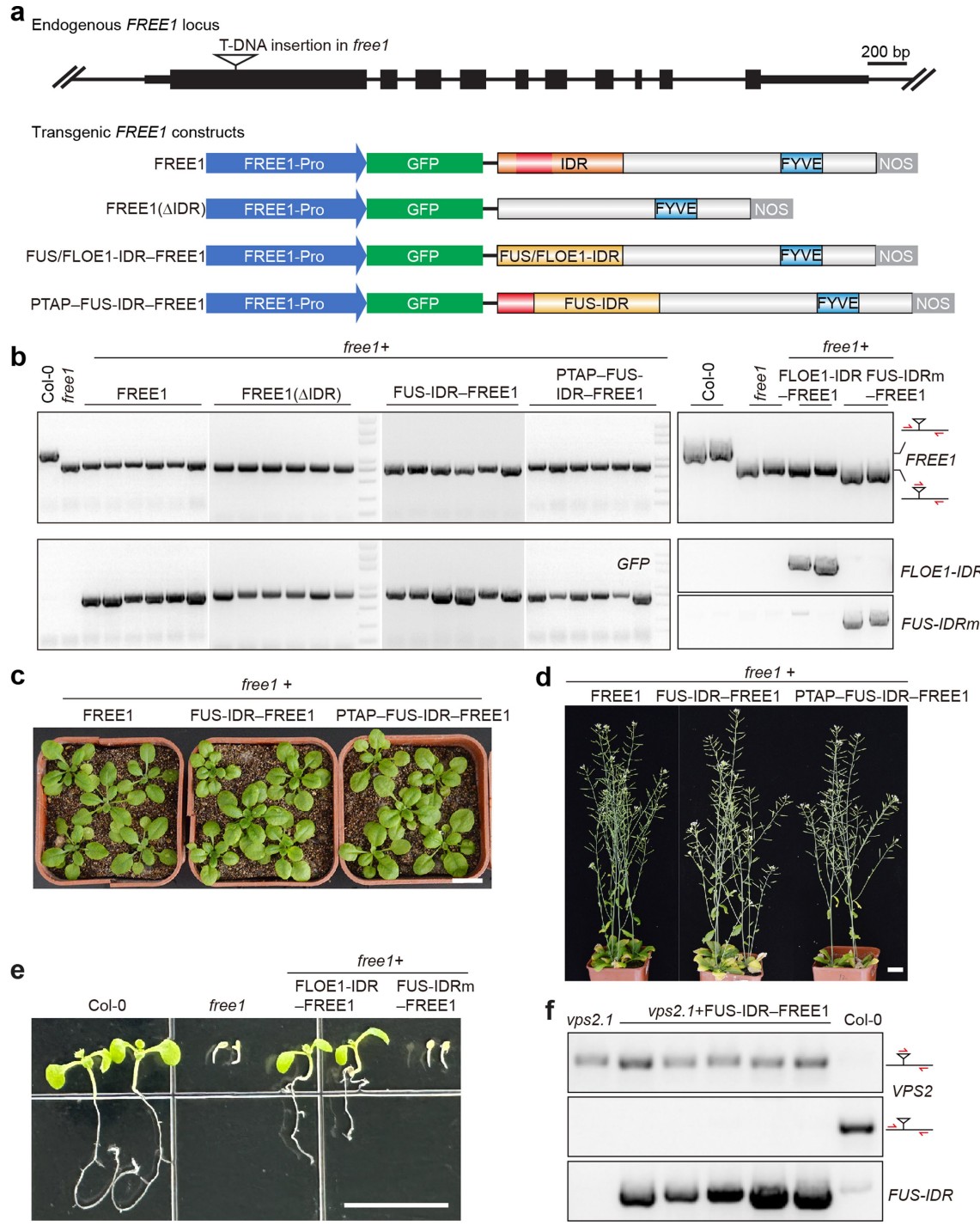

**Extended Data Fig. 3 | Characterization of transgenic plants expressing chimeric FREE1 variants. a**, Illustration of the genomic *FREE1* locus (top panel) and transgenic *FREE1* constructs (bottom panel). Thick black boxes indicate exons, thin black boxes indicate UTRs and black lines indicate introns. Primers for genotyping are indicated by red arrows. **b,f**, Electrophoresis of PCR genotyping products using primers as indicated. **c,d**, Photographs of 3-week-old (c) and 6-week-old (d) plants of indicated genotypes. Scale bars, 2 cm. **e**, Seven-day-old *A. thaliana* seedlings of indicated genotypes. Scale bar, 1 cm. Representative images of *n* independent experiments (*n* = 2 (b,f), *n* = 3 (c-e)). For b and f, the unprocessed gels are provided in SI Fig. 1.

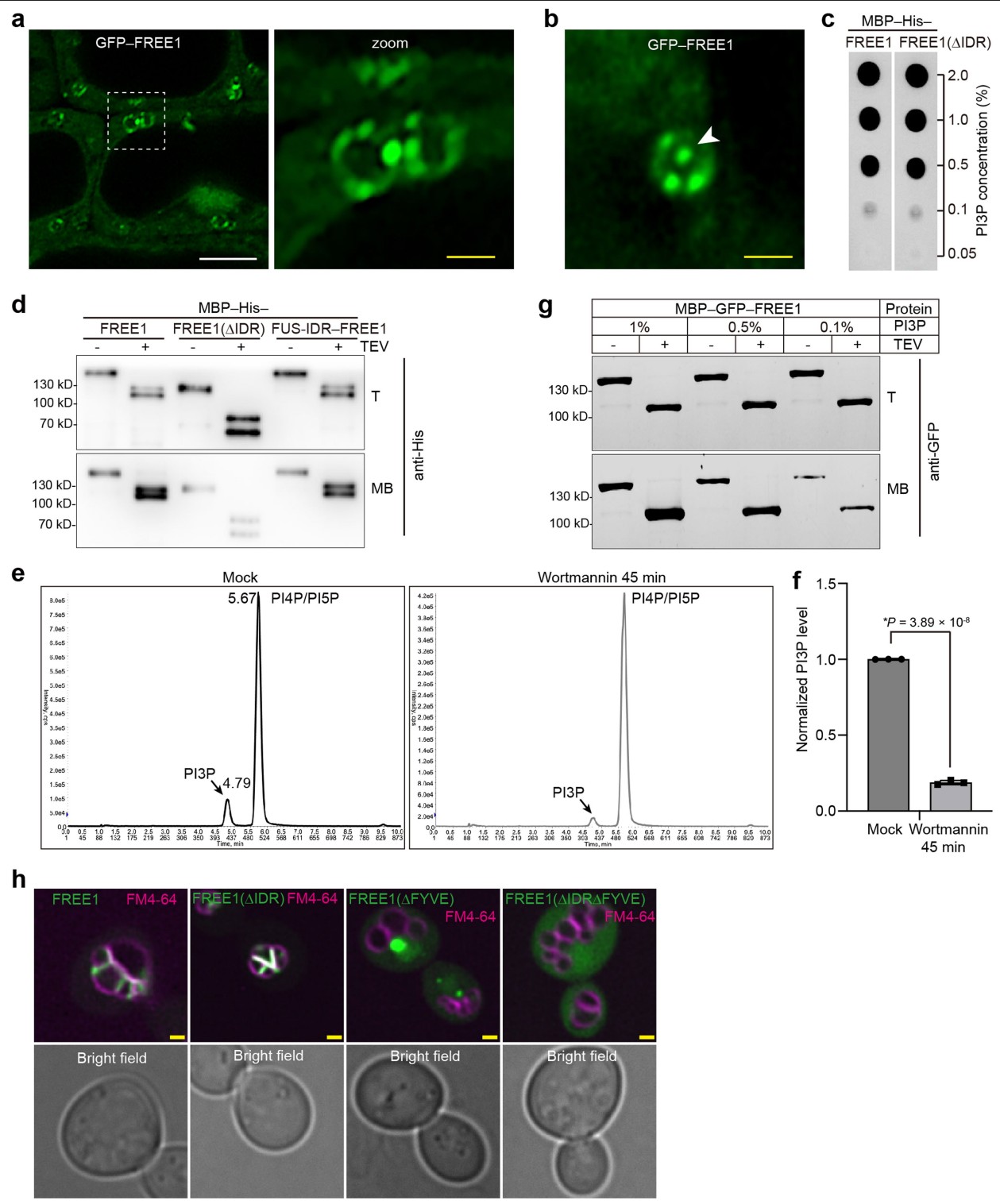

**Extended Data Fig. 4 | FREE1 condensates localise to PI3P-containing membranes in vivo and in vitro. a**, **b**, Confocal microscopy image showing the localization of FREE1 on the surface (a) and within (b) wortmannin-enlarged MVBs in *A. thaliana* root tip cells. **c**, Dot blot assay showing the binding of condensing and non-condensing FREE1 to PI3P. **d**, **g**, Immunoblot of total (T) and membrane-bound (MB) proteins. TEV protease cleavage of the solubility tag induces LLPS. **e**, Ion chromatogram (XIC) of PI3P peak at Rt = 4.79 min, separated from PI4P or PI5P peak at Rt = 5.67 min obtained from injecting indicated lipid extract (details of sample preparation is discussed in methods).

**f**, Normalised PI3P levels. Error bars indicate mean ± SD. Wortmannin treatment reduced PI3P levels to 18.8% ± 1.4%. *P* values are indicated (two-tailed *t* test, *n* = 3 experiments). Asterisks indicate significant differences. **h**, Expression of FREE1 variants in *Saccharomyces cerevisiae*. Indicated variants of GFP–FREE1 (green) were expressed under the control of the GAP1 promoter and observed by confocal fluorescence microscopy (upper panels). Prior to visualization, yeast vacuolar membranes were stained with FM4-64 dye (magenta). Scale bars, 5 μm (white), 1 μm (yellow). Representative images of *n* independent experiments (*n* = 3 (a-e, g, h)). For c, d and g, the unprocessed blots are provided in SI Fig. 1.

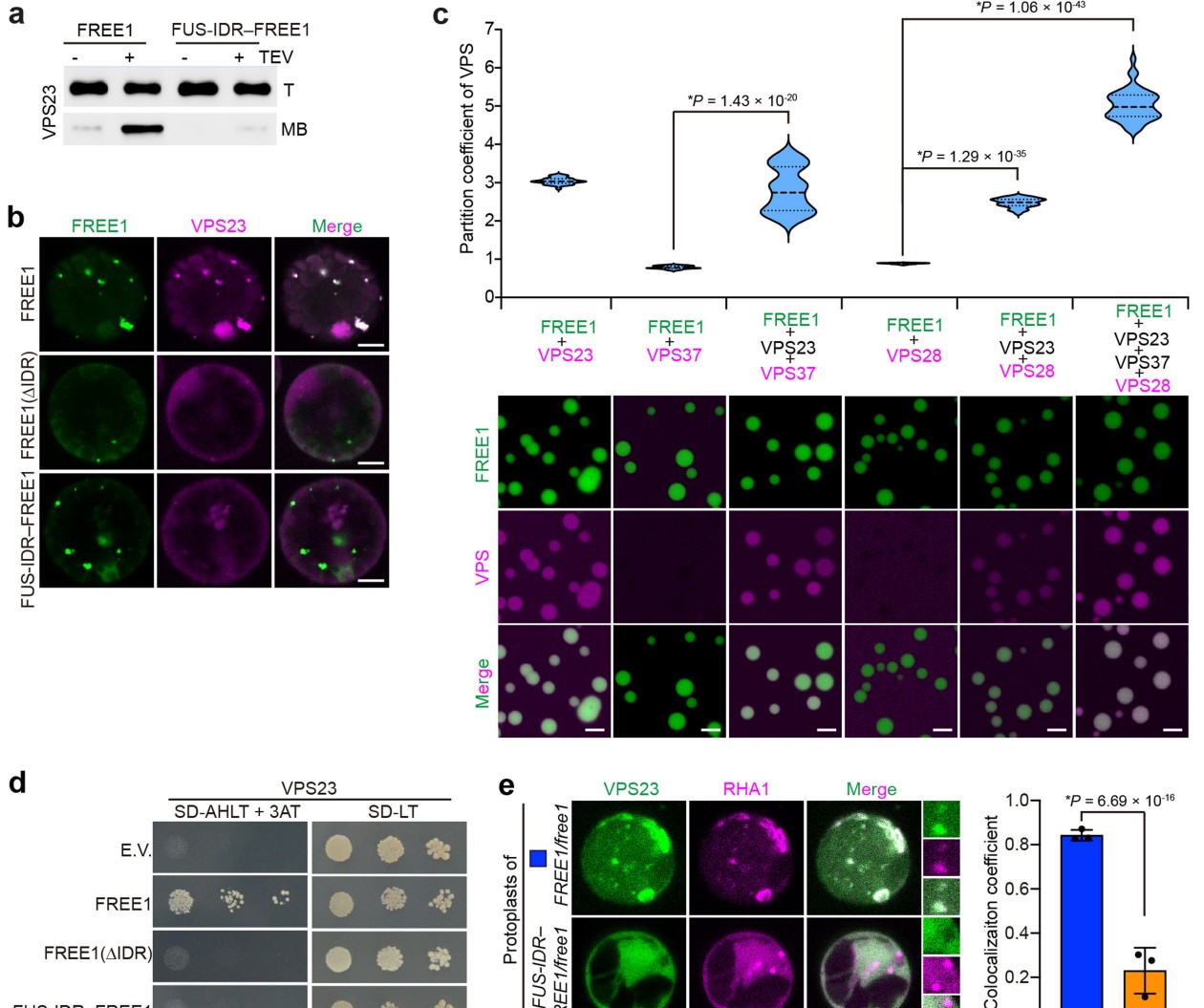

**Extended Data Fig. 5 | Phase-separated FREE1 condensates partition ESCRT-I subunits. a**, Immunoblot of total (T) and membrane-bound (MB) VPS23. TEV protease cleavage of the MBP solubility tag induces LLPS (see Fig. 2e). **b**, Confocal microscopy images of *A. thaliana* protoplast cells co-expressing VPS23 with FREE1 variants (see Fig. 2g). **c**, Bottom panel, confocal microscopy images showing the enrichment of purified VPS proteins (magenta font and image, with C-terminal mCherry tags) by partitioning to FREE1 condensates (green font and image). Unlabelled VPS proteins (black) were added as indicated. Top panel, quantification of 561-nm emission (magenta) inside and outside of FREE1 condensates. Black dashed lines indicate median. *P* values are indicated

(two-tailed *t* test, *n* = 32, 25, 34, 24, 28, 42 condensates, respectively). Asterisks indicate significant differences. **d**, Interactions between FREE1 variants and VPS23 as determined by yeast two hybrid assay. E.V., empty vector. **e**, Left, confocal microscopy images of indicated protoplast cells co-expressing VPS23 with RHA1. Right, quantification of the colocalization coefficient between VPS23 and RHA1. Error bars indicate mean ± SD. *P* values are indicated (two-tailed *t* test, *n* = 39, 38 cells, respectively). Asterisk indicates significant difference. Scale bars, 5 μm. Representative images of *n* independent experiments. (*n* = 3 (a-e)). For a, the unprocessed blots are provided in SI Fig. 1.

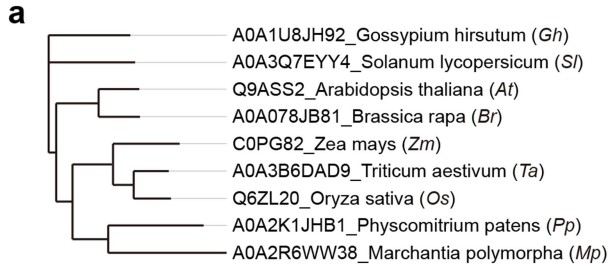

**a**

A0A1U8JH92_Gossypium hirsutum (*Gh*)
A0A3Q7EYY4_Solanum lycopersicum (*Sl*)
Q9ASS2_Arabidopsis thaliana (*At*)
A0A078JB81_Brassica rapa (*Br*)
C0PG82_Zea mays (*Zm*)
A0A3B6DAD9_Triticum aestivum (*Ta*)
Q6ZL20_Oryza sativa (*Os*)
A0A2K1JHB1_Physcomitrium patens (*Pp*)
A0A2R6WW38_Marchantia polymorpha (*Mp*)

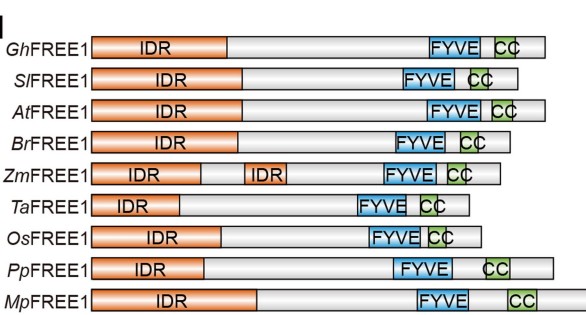

**b**

**c**

| Protein | Fold enrichment by b-isox | Reference |
|---------|---------------------------|-----------|
| *Gh*FREE1 | 181 | *unpublished* |
| *Sl*FREE1 | 614 | ref.17 |
| *At*FREE1 | 1858 | ref.17 |
| *Br*FREE1 | 724 | ref.17 |
| *Zm*FREE1 | 1963 | ref.17 |
| *Ta*FREE1 | 2324 | ref.17 |
| *Os*FREE1 | 594 | ref.17 |
| *Pp*FREE1 | 165 | ref.17 |
| *Mp*FREE1 | 215 | *unpublished* |

**Extended Data Fig. 6 | FREE1 is conserved in land plants. a**, Phylogenetic tree of FREE1 homologs from indicated species. **b**, Multiple sequence alignment of FREE1 homologs from indicated species. **c**, A table showing the enrichment of FREE1 homologs by b-isox. **d**, Domain structure of FREE1 homologs. IDRs were predicted by PONDR.

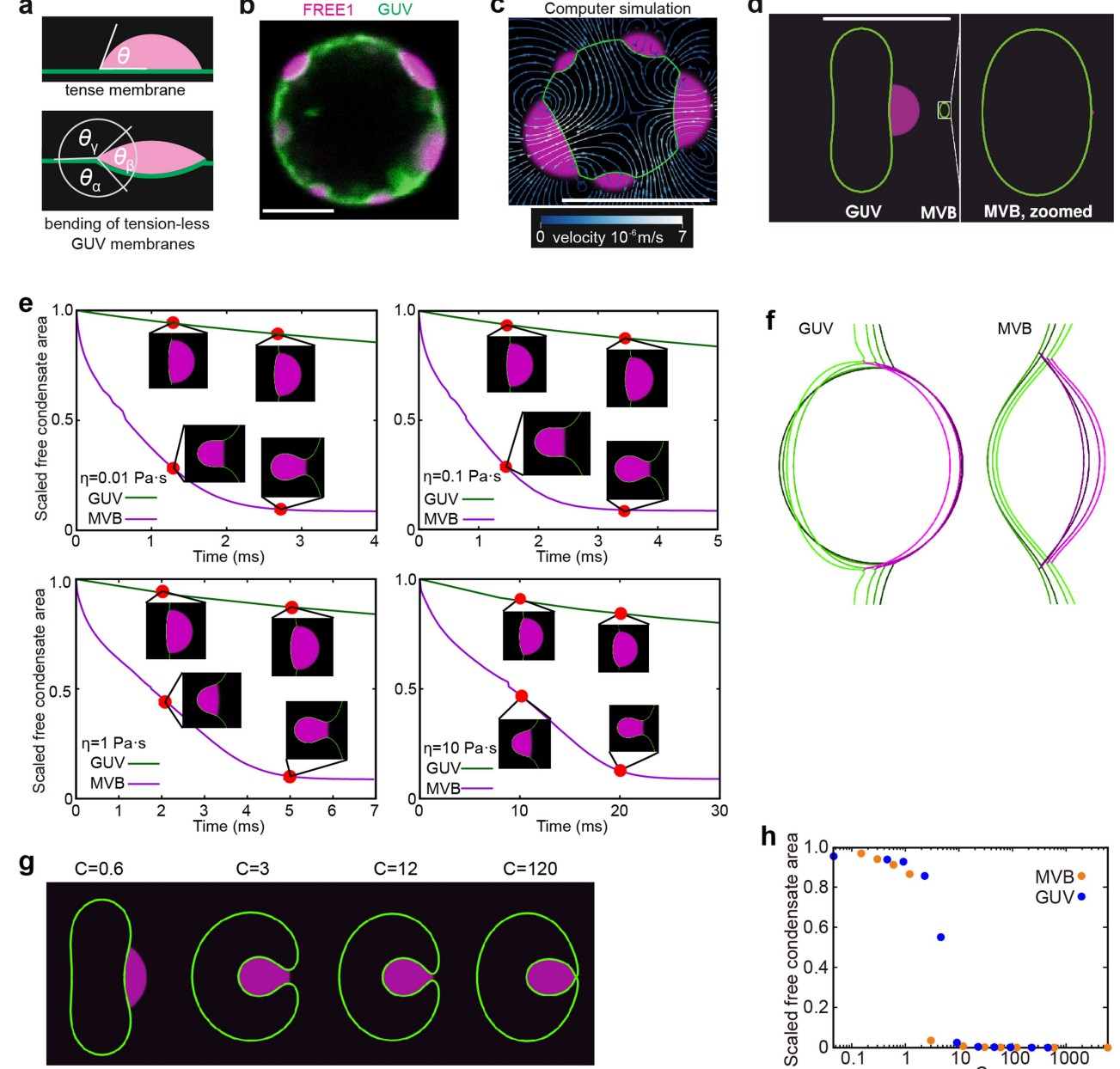

**Extended Data Fig. 7 | Computer simulations of condensate-membrane wetting and ILV formation. a**, Capillary forces deform tension-free membranes. θ was computed via three contact angles of tension-less GUVs. **b**, FREE1 condensates (magenta) wet GUV membranes (green). These data were used to determine the wetting contact angle θ = 70° ± 20° (n = 11 condensates). **c**, Dynamic remodelling of GUV wetted by several 1–3 μm sized condensates assuming θ = 70°. Two-dimensional computer simulation allowed modelling the wetting of multiple condensates on a single GUV. Membrane tension σ_αγ = 1.5 mN/m, membrane tension σ_αβ = surface tension σ_βγ = 1.1 mN/m, viscosity η = 10 Pas. Streamlines coloured by velocity magnitude indicate capillary force-induced fluid flow, which the membrane follows consistently (there is no flow through the membrane). **d**, Initial state of axisymmetric 3D simulations to assess ILV formation dynamics. Condensate half circles were placed onto bending energy minimizing MVB and GUV shapes, which were determined by running condensate-free simulations. See also Supplementary Videos 1 & 2. **e**, Dynamics of condensate-mediated ILV formation in MVBs (magenta lines) and GUVs (green lines) for varying ambient viscosities η_ambient = 0.01-10 Pas. Insets show intermediate morphologies as indicated. Scaled free condensate areas (ratio of condensate/fluid surface with respect to initial value) decrease over time. θ = 70°, membrane tension σ_αγ = 1.5 mN/m, membrane tension σ_αβ = surface tension σ_βγ = 1.1 mN/m, condensate viscosity η_cond = 10 Pas, ambient (cytosol or buffer) viscosity η_ambient = 0.01-10 Pas. Condensate diameters are 2 μm and 35 nm in the GUV-like and MVB-like simulations, respectively. Obtained stationary neck radius is 6.5 nm for MVB. **f**, 3D simulation snapshots comparing membrane (green) and condensate (magenta) surface shapes at free condensate area = 0.5 and η_cond/η_ambient = 1 (darkest) to η_cond/η_ambient = 1000 (lightest). **g**, Stationary 3D shapes for wetting simulations of a large, single condensate (2 μm) and a GUV with increasing value of the parameter C = σ_βγD_ILV²/κ. **h**, Scaled free condensate area (ratio of condensate/fluid surface with respect to its initial value) demonstrates dependence of condensate invagination on C. Transition occurs at C≈1 for small MVB-sized (orange) and large GUV-sized (blue) condensates.

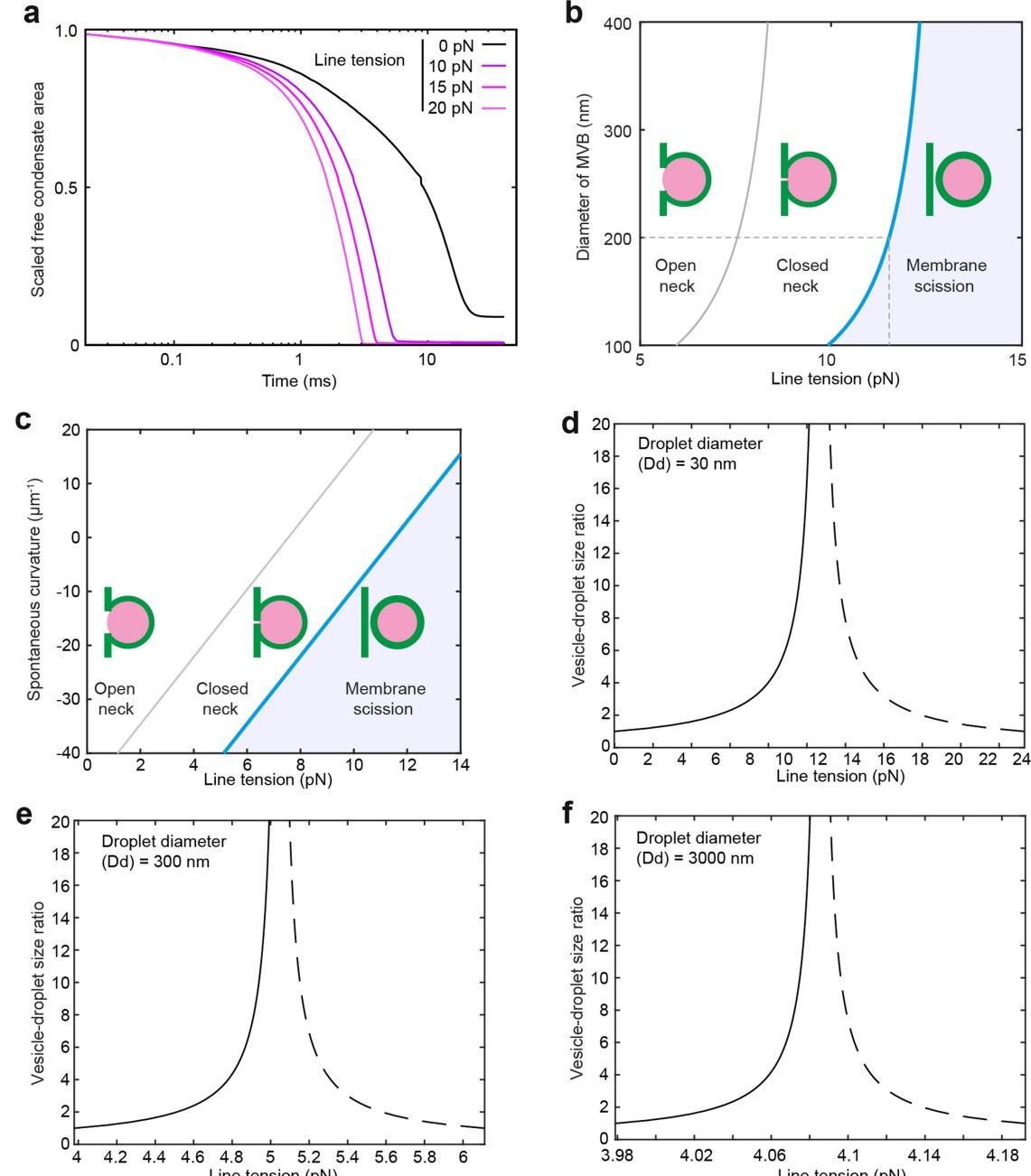

**Extended Data Fig. 8 | Line tension increases neck formation dynamics and induces membrane scission. a**, Dynamics of ILV formation for varying line tensions. Scaled free condensate area (ratio of condensate/fluid surface with respect to initial value) decreases over time. $\theta = 70°$, membrane tension $\sigma_{\alpha\gamma} = 1.5$ mN/m, membrane tension $\sigma_{\alpha\beta}$ = surface tension $\sigma_{\beta\gamma} = 1.1$ mN/m. Viscosities $\eta_{cond} = 10$ Pas, $\eta_{ambient} = 0.01$-10 Pas. Condensate diameter is 35 nm. **b**, Theoretical stability diagram of condensate-induced membrane scission for varying MVB size. Blue line, critical scission line tension $\lambda^*$. Critical neck

constriction force $f^* = 25$ pN, bending rigidity $\kappa = 10^{-19}$ J, $D_{ILV} = 35$ nm. Grey dashed line indicates $D_{MVB} = 200$ nm. **c**, Theoretical stability diagram of condensate-induced membrane scission and varying membrane spontaneous curvature. Blue line, critical scission line tension $\lambda^*$. Critical neck constriction force $f^* = 25$ pN, bending rigidity $\kappa = 10^{-19}$ J, $D_{MVB} = 200$ nm, $D_{ILV} = 35$ nm. **d-f**, The line tensions required for scission for inward budding (solid line) and outward budding (dashed line) at three physiologically relevant condensate sizes (30 nm, 300 nm and 3000 nm) and a range of relative membrane sizes.

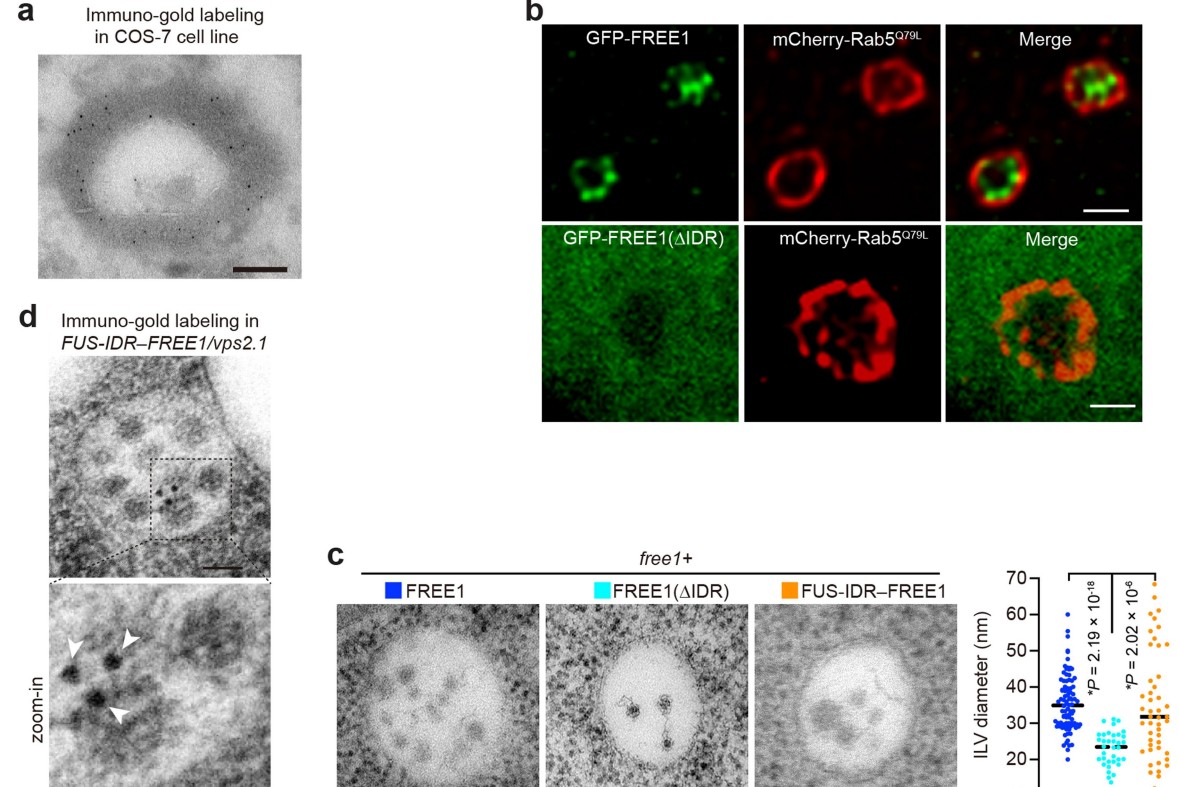

**Extended Data Fig. 9 | Phase separated FREE1 condensates wet membranes and form ILVs in cells. a**, Representative electron microscopy images of immunogold-labelled samples using GFP antibodies in COS-7 cells expressing GFP–FREE1. **b**, Mammalian COS-7 cells co-expressing full-length FREE1 or FREE1(ΔIDR) with Rab5$^{Q79L}$ were fixed and imaged using a super resolution (structured illumination) microscope. **c**, Left, TEM analysis of MVBs in *A. thaliana* root cells of the indicated genotypes. Right, quantification of the diameter of ILVs. $n$ = 90, 36, 47 MVBs, respectively. Error bars indicate mean ± SD. Asterisks indicate significant differences ($P$ values are indicated, two-tailed $t$ tests). **d**, Imaging of immuno-gold labelled of GFP–FUS-IDR–FREE1 in *A. thaliana vps2.1* knock-out embryo cells. Representative images of $n$ independent experiments. ($n$ = 3 (a-d)). Scale bars, 1 μm (white), 100 nm (black).

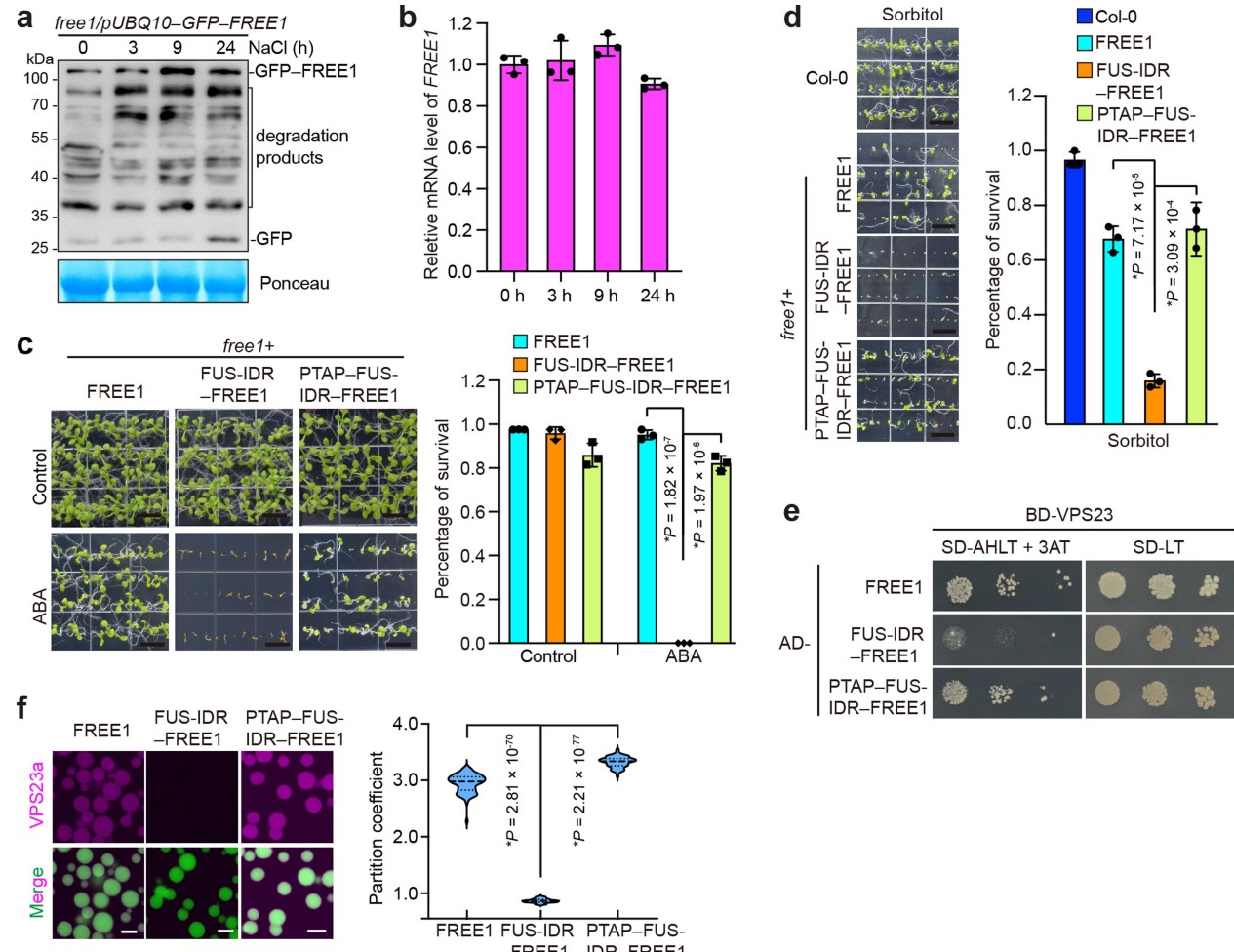

**Extended Data Fig. 10 | The role of FREE1 condensation during osmotic stress. a,b**, Accumulation of FREE1 protein (a) and mRNA (b) upon NaCl treatment. **c,d**, *A. thaliana* seedlings subjected to abscisic acid (c) and osmotic (d) treatments. Left panels, images of ten-day-old *A. thaliana* plants. Right panels, individual survival rates (with mean ± SD). *P* values are indicated (two-tailed *t* test, *n* = 3 experiments). Asterisks indicate significant differences. **e**, Interactions between FREE1 variants and VPS23 as determined by the yeast two hybrid assay. **f**, VPS23 partitioning in condensates of FREE1 variants. Left, confocal images. Right, quantification of partitioning. Error bars indicate mean ± SD. *P* values are indicated (two-tailed *t* test, *n* = 30 condensates). Asterisks indicate significant differences. Representative images of *n* independent experiments (*n* = 2 (a, e), *n* = 3 (b)). Scale bars, 5 μm (white), 1 cm (black). For a, the unprocessed blots are provided in SI Fig. 1.

# Reporting Summary

## Statistics

For all statistical analyses, confirm that the following items are present in the figure legend, table legend, main text, or Methods section.

| n/a | Confirmed | |
|---|---|---|
| ☐ | ☒ | The exact sample size (*n*) for each experimental group/condition, given as a discrete number and unit of measurement |
| ☐ | ☒ | A statement on whether measurements were taken from distinct samples or whether the same sample was measured repeatedly |
| ☐ | ☒ | The statistical test(s) used AND whether they are one- or two-sided *Only common tests should be described solely by name; describe more complex techniques in the Methods section.* |
| ☒ | ☐ | A description of all covariates tested |
| ☒ | ☐ | A description of any assumptions or corrections, such as tests of normality and adjustment for multiple comparisons |
| ☐ | ☒ | A full description of the statistical parameters including central tendency (e.g. means) or other basic estimates (e.g. regression coefficient) AND variation (e.g. standard deviation) or associated estimates of uncertainty (e.g. confidence intervals) |
| ☐ | ☒ | For null hypothesis testing, the test statistic (e.g. $F$, $t$, $r$) with confidence intervals, effect sizes, degrees of freedom and $P$ value noted *Give P values as exact values whenever suitable.* |
| ☒ | ☐ | For Bayesian analysis, information on the choice of priors and Markov chain Monte Carlo settings |
| ☒ | ☐ | For hierarchical and complex designs, identification of the appropriate level for tests and full reporting of outcomes |
| ☒ | ☐ | Estimates of effect sizes (e.g. Cohen's *d*, Pearson's *r*), indicating how they were calculated |

*Our web collection on statistics for biologists contains articles on many of the points above.*

## Software and code

Policy information about availability of computer code

| Data collection | All fluorescence imaging data of plant cells were collected on Zeiss LSM880 confocal microscope and Olympus Fluoview FV-1000 confocal laser microscope; ; The super resolution microscopy data was collected using a Nikon combined confocal A1/SIM/STORM system; Imaging of yeast cells was performed on SpinSR10 spinning disc confocal microscope; Transmission electron microscopy data of plant samples was collected using a 80kV Hitachi H-7650 transmission electron microscope (Hitachi High-Technologies Corporation, Japan); Transmission electron microscopy data of mammalian cells was collected using a HT-7800 120 kV transmission electron microscope (Hitachi High-Technologies); Immuno detection of proteins was performed by chemiluminescence (ChemiDoc, LAS4000). |
|---|---|
| Data analysis | Data analysis was performed in the Fiji implementation of ImageJ (Version 1.51), Cellsens software (Olympus, Japan, Version 4.2), NIS-Elements (Nikon) software (Version Ar), ZEN black software (version 2.3) |

For manuscripts utilizing custom algorithms or software that are central to the research but not yet described in published literature, software must be made available to editors and reviewers. We strongly encourage code deposition in a community repository (e.g. GitHub). See the Nature Portfolio guidelines for submitting code & software for further information.

## Data

Policy information about [availability of data](availability of data)

 All manuscripts must include a [data availability statement](data availability statement). This statement should provide the following information, where applicable:

- Accession codes, unique identifiers, or web links for publicly available datasets
- A description of any restrictions on data availability
- For clinical datasets or third party data, please ensure that the statement adheres to our [policy](policy)

Data availability
All data are available in the main text or the supplementary materials. Full version of all gels and blots are provided in Supplementary Figure 1. Source data are provided with this paper. Figures describing the results of the model can be reconstructed directly from the equations and procedures presented in the paper and in the Supplementary Methods, and require no data.

Code availability
The numerical simulation code is available at [cite: https://doi.org/10.5281/zenodo.11919014]. Installation requires the finite element library AMDiS, which can be downloaded at [https://gitlab.math.tu-dresden.de/iwr/amdis]

## Research involving human participants, their data, or biological material

Policy information about studies with [human participants or human data](human participants or human data). See also policy information about [sex, gender (identity/presentation), and sexual orientation](sex, gender identity/presentation, and sexual orientation) and [race, ethnicity and racism](race, ethnicity and racism).

| | |
|---|---|
| Reporting on sex and gender | N/A |
| Reporting on race, ethnicity, or other socially relevant groupings | N/A |
| Population characteristics | N/A |
| Recruitment | N/A |
| Ethics oversight | N/A |

Note that full information on the approval of the study protocol must also be provided in the manuscript.

# Field-specific reporting

Please select the one below that is the best fit for your research. If you are not sure, read the appropriate sections before making your selection.

☒ Life sciences        ☐ Behavioural & social sciences        ☐ Ecological, evolutionary & environmental sciences

For a reference copy of the document with all sections, see [nature.com/documents/nr-reporting-summary-flat.pdf](nature.com/documents/nr-reporting-summary-flat.pdf)

# Life sciences study design

All studies must disclose on these points even when the disclosure is negative.

| | |
|---|---|
| Sample size | Sample sizes are chosen as widely used in the field, for example, Fang et al., Nature 569: 265-269. Biological and technical replicates were performed as described in the Methods for each experiment and conform to standards in the field. Exact n numbers for each experiment are provided in each figure legend. |
| Data exclusions | No data was excluded from analysis. |
| Replication | All attempts at replication were successful. All experiments were repeated on different days at least twice. The sample number and biological replicate number are indicated in the legends. |
| Randomization | Randomization of samples were performed. Seedlings from different plates were collected. |
| Blinding | Blinding was not deemed necessary in our study since we made no a prior assumptions on the response of the different samples to the experimental treatment, samples were all treated in parallel and all samples treated were always measured. |

# Reporting for specific materials, systems and methods

We require information from authors about some types of materials, experimental systems and methods used in many studies. Here, indicate whether each material, system or method listed is relevant to your study. If you are not sure if a list item applies to your research, read the appropriate section before selecting a response.

## Materials & experimental systems

| n/a | Involved in the study |
|-----|----------------------|
| ☐ | ☒ Antibodies |
| ☐ | ☒ Eukaryotic cell lines |
| ☒ | ☐ Palaeontology and archaeology |
| ☒ | ☐ Animals and other organisms |
| ☒ | ☐ Clinical data |
| ☒ | ☐ Dual use research of concern |
| ☐ | ☒ Plants |

## Methods

| n/a | Involved in the study |
|-----|----------------------|
| ☒ | ☐ ChIP-seq |
| ☒ | ☐ Flow cytometry |
| ☒ | ☐ MRI-based neuroimaging |

## Antibodies

**Antibodies used**

GFP (Roche, 11814460001; 1:7000), FLAG (Merck, F1804; 1:2000), tubulin (Sigma, T5168; 1:2000), ubiquitin (Santa Cruz Biotechnology, sc-8017; 1:1000), His-tag (Sangon, D110002; 1:1000), the horseradish peroxidase (HRP)-conjugated secondary antibodies Goat anti-Mouse (CWBIO, CW0102; 1:10000) and Goat anti-Rabbit (CWBIO, CW0103; 1:10000).

**Validation**

GFP (Roche, 11814460001): https://elabdoc-prod.roche.com/LifeScience/Document/27e718d4-c7c2-e711-b48d-00215a9b3428
FLAG (Merck, F1804): https://www.sigmaaldrich.com/certificates/sapfs/PROD/sap/certificate_pdfs/COFA/Q14/
F3165-1MG0000350182.pdf
tubulin (Sigma, T5168): https://www.sigmaaldrich.com/certificates/sapfs/PROD/sap/certificate_pdfs/COFA/Q14/
T5168-100UL0000359065.pdf
His-tag (Sangon, D110002): https://store.sangon.com/productImage/DOC/D110002/D110002_EN_D.pdf
ubiquitin (Santa Cruz Biotechnology, sc-8017; 1:1000): https://datasheets.scbt.com/sc-8017.pdf

## Eukaryotic cell lines

Policy information about cell lines and Sex and Gender in Research

**Cell line source(s)**

COS-7 cells are kind gift from Dr. Yiguo wang (Tsinghua University); SEY6210 WT Saccharomyces cerevisiae cells: Robinson et al. (1988) Mol Cell Biol 8(11):4936-48

**Authentication**

None of the cell line used have been authenticated.

**Mycoplasma contamination**

Cells tested negative for mycoplasma contamination.

**Commonly misidentified lines**
(See ICLAC register)

No commonly misidentified cell lines were used.

## Plants

**Seed stocks**

The free1 mutant (a transposon insertion Arabidopsis mutant line 15-1960-1 from RIKEN) seeds and UBQ10::GFP-FREE1 transgenic seeds were kindly provided by Prof. Liwen Jiang (The Chinese University of Hong Kong, China); The vps2 mutant (a T-DNA insertion line GABI_670D06) seeds were kindly provided by Prof. Youfa Cheng (Chinese Academy of Sciences, China)

**Novel plant genotypes**

All other transgenic A. thaliana lines (FREE1 and FREE1 variants, FREE1ΔIDR, iDRFUS-FREE1, PTAP-iDRFUS-FREE1, iDRFUSm-FREE1 and IDRFLOE1-FREE1) were generated by Agrobacterium transformation of corresponding constructs into the heterozygous free1 mutant background and subsequent genotyping of homozygous free1 from progeny.

**Authentication**

Primers used for genotyping T-DNA insertion lines are provided in Supplementary Table 1 in the methods section.

