## [Peer Review File · Nature]

Manuscript Title: Biomolecular condensates mediate bending and scission of endosome membranes

Reviewer Comments & Author Rebuttals

Reviewer Reports on the Initial Version:

Referees' comments:

Referee #1 (Remarks to the Author):

This paper presents evidence that a membrane-binding IDP (FREE1), a component of the ESCRT-III machinery, is capable of driving internalization of vesicles into multi-vesicular bodies *in vivo*, without the complete ESCRT-III machinery. Furthermore, they show that the mechanism has a clear phenotype *in vivo*.

The result is significant and of general interest because of the high level of interest in protein condensates and to what degree they matter.

Here is a result where they appear to matter: the combination of their membrane-binding and condensate-water tension is a concrete physical model of how they bind surfaces.

Results: FREE1 is enriched in isolates driven by b-isox. Of the enriched proteins, only FREE1 forms condensates by itself.

When the intrinsically disordered region (IDR) is removed, no condensate is formed.

The delta-IDR form is then used as a test of when FREE1-driven condensation is operative *in vivo*.

Delta-IDR does not rescue a previously known lethal FREE1 knockout.

An alternative (FUS) condensing IDR substitution of the FREE1 IDR rescues knockout.

An ESCRT-III protein (VPS23) does not interact with IDR/FUS-FREE1, suggesting that the FUS-based rescue is independent of the entire ESCRT-III machinery.

The authors highlight the possibility that the entire ESCRT-III complex is not required for *in vivo* scission.

It is challenging to show that any particular molecular mechanism is operative *in vivo*. For example, the entire ESCRT-III machinery cannot be removed to show that MVBs can be formed using FREE1 alone, or that scission is possible without ESCRT-III.

The study here relies on 1) reconstitution of internalization including scission *in vivo*, 2) the observation that FUS-IDR recombinant proteins don't interact with an element of ESCRT-III but can rescue FREE1 knockout, and 3) modeling.

I am convinced that condensation is a requirement for a critical stage of vesicle internalization analogous to coat formation by clathrin or COPII.

However, I don't see how this paper can exclude that scission is still occurring *in vivo* by extended ESCRT machinery.

The extended ESCRT machinery likely has affinity for fission necks, where it performs its action and

to which its scaffolding shape is matched.

While FREE1 clearly has an impact on membrane localization of VPS23 (Fig 2d), it is not clear that it is as decisive as the paper implies ("associates with SUVs only when FREE1 is present") either in vitro or in vivo.

My expertise is modeling.

The paper supplements the experiments with two models.

One model is a model with origins from the engineering field, with results shown in 3c and 3d.

This paper applies unique expertise and methodology to measure the time-dependence of membrane reshaping (Fig 3d) incorporating surface tensions and viscosities.

This is an intriguing possibility but there is a major issue translating engineering problems to membrane physics.

From an engineering perspective, the membrane deformation is characterized by a strain, with forces determined by a matrix of force constant terms that modulate pairs of strains.

Membrane biophysics has taken a somewhat different path to the Helfrich/Canham Hamiltonian, mainly because the surface is a liquid crystal that cannot support a lateral shear strain.

Yet the neck at 30 milliseconds is a shape that I believe can be identified to be modeled by a surface that penalizes lateral shear strain.

(The authors write in the supplemental that a shear modulus (K_s) is applied).

Instead, membrane necks are very small, where saddle curvature is penalized by a Gaussian curvature modulus that cannot clearly be modeled by engineering moduli.

Four other minor comments on the model:

First, was the curved shape shown in Fig 3d chosen as the zero strain reference state of the membrane, or is the zero strain shape flat? The zero strain reference state of a vesicle should be flat, and this should result in a sphere as the minimum energy state, not something oblong.

Second, it appears a uniform viscosity was chosen. Yet water has a viscosity one thousand times smaller than a condensate. How would a correct specification of water viscosity affect kinetics?

Third, the authors must be clearer about the dimensionality of the system.

It is very hard to tell if this work uses three dimensional modeling, if it uses two dimensional modeling using axial symmetry, or if it uses purely 2 dimensional modeling with the system isotropic "out of the page".

Fourth, what are the flows in Fig 3c. Is something flowing through the bilayer? If so, this shouldn't be. If not, to me it's very confusing.

While I am very much excited by the model, I did not see its important connection to the main result of the paper, especially including my confusion about the points above.

The second model is a traditional membrane biophysics model applying standard modeling constants based on theory from the Lipowsky group.

I strongly warn the authors against over-interpreting the energetics of leaflets using continuum theory near the pinching point (Supplemental Eqs. 5 and 6).

There are three related issues:

One, for a very small exo- or endo-cytic pore, while the mean curvature of the leaflets might be zero at the bilayer midplane, the curvature of the inner (collapsing) leaflet diverges. This difference between leaflet curvature and bilayer curvature only applies at very high curvatures (exactly this

scenario).

From Agudo-Canalejo and Lipowsky, 2016, the source of the equations from which the authors derive subsequent results:

"Predicting whether the closed neck will break, leading to fission, or whether it will remain intact, leaving the topology of the membrane unchanged, lies beyond the realm of applicability for the continuum membrane model used here. As an example, our continuum model allows for the formation of infinitely narrow membrane necks, whereas the diameter of a real neck must exceed twice the membrane thickness, [approximately] 4–5 nm."

Two, for topological changes of the membrane the sum Gaussian curvature changes discontinuously, and we currently don't know where this model energy comes from or goes (see Siegel and Kozlov 2004 Biophys J 87 366 Fig. 2 for an illustration and the Appendix for how to analyze the leaflets).

Three, proteins simply may not fit around a very small pore, especially in the endocytic case. Yet their effect at this critical site appears to still be included in the theory.

The sum of these observations is that we really don't know what's going on for very small pores. We do know that the cell typically brings in extremely powerful actors like dynamin on the outside to complete fission.

Not only is the scission theory likely invalid (engulfment is fine), but scission may still be a transition state with a significant activation energy and still be consistent with their experiments.

I believe the process is effectively irreversible after the final interior-coated particle is made.

The authors' proof-of-principle theory would be far more likely to stand the test of time if, instead of a spontaneity relation for full scission, the authors used their theory to place the scission-inducing line tension in the context of the line tension required to stabilize incomplete internalization. I think this may be done with minor shifts in language in the main text and severe caveats placed on the applicability of elasticity theory to fission.

I am unsure if readers will understand the connection to spontaneous curvature.. I think the model would be just as strong focusing clearly on surface tensions giving rise to line tension and mentioning that protein-lipid associations change spontaneous curvature, likely supporting fusion and lowering the requirement of a strong line tension.

The narrative of the article was very well put together, with the authors frequently anticipating my next thought immediately with a well described experiment. In my view I would advise strongly against any revision of the narrative structure.

In summary, I am very excited about this paper and found the set of experiments to be convincing about the necessity of phase separation but only suggestive about the simplicity of the scission mechanism, based both on the nearly insurmountable ambiguity of in vivo experiments and the failure of membrane continuum theory to address scission.

The experiments have not dramatically shifted my thinking about the necessity of complicated scission machinery in vivo, but they have certainly opened the possibilities. It has continued to shift my thinking about the necessity of complicated coats (like clathrin).

Put in the context of what I believe is a very exciting and challenging frontier of experiment and theory, this paper is a critical contribution.

In many places, the authors write carefully to indicate that the condensate-driven scission mechanism is only suggested.

The two places where I disagreed with language were the applicability of continuum theory to scission, and the exclusion of the full ESCRT-III machinery from scission by the weak association of VPS23 and FUS-IDR modified FREE1 as proof that VPS23-associated elements would not participate in scission.

Referee #2 (Remarks to the Author):

The major experimental result of the study is that the plant ESCRT component, FREE1, forms liquid condensate. The FREE1 condensate droplets binds and can be engulfed by lipid bilayers of GUV. Moreover a membrane neck connecting an engulfed droplet with the initial GUV membrane can undergo fission. In addition, FREE1 condensation was observed also in cells.

The major message of the study is that droplets of FREE1 condensate droplets can mediate membrane budding and fission in the process of formation of internal vesicles of MVBs.

In addition, theoretical modeling is presented with a goal to support the observations.

I have the following reservations concerning this work.

1. The major proposal concerning the intracellular membrane budding/fission by FREE1 liquid droplets remains a speculation. The authors should demonstrate existence within cells (MVBs) of FREE1 droplets covered by membranes. Demonstration of such covered droplets in GUVs is not sufficient.

2. The whole theoretical part of the article does not advance the understanding. It is not new in its essence on one hand, and does not provide any new insight into the system, on the other. In addition, it is confusing.

Specifically,

- lines 183-186. This criterion of a particle engulfment by a membrane has been derived in multiple works and does not need any complex consideration of the system dynamics.

In general the simulations of the dynamics (Fig.3C) are not justified by the experimental results and are, therefore, unnecessary here.

- The purpose of the computations of the shapes (Fig.3d) is unclear. Such calculation were done in multiple previous works so that the present one does not provide any new understanding.

- The fission phase diagram is very speculative. The used values of the model parameters are not well justified. In particular,

(i) the values of the tensions are estimated from the relationship (Eq.8) of the Supplement, which implies the membrane under the droplet to be flat, which is not the case.

(ii) The origin of the line tension is obscure. I cannot think of any reasonable microscopic origin of the line tension in this particular system. A reference to a general property of a three-phase boundary is insufficient to justify the existence of the line tension not to mention an estimation of its value.

(iii) The criterion of membrane fission used here is also rather unconvincing. The authors use the previously published relationships (Eqs. 1-6) of the second part of the Supplement, which were derived for tension-free membranes, while in the current work, the membranes (at least one of them if not both of them) are under considerable tension. It is easy to see that the presence of tension qualitatively changes the expression of the neck expanding force.

Referee #3 (Remarks to the Author):

Feng et al. present compelling *in vivo* and *in vitro* data to support a model for ESCRT- and ATP-independent formation of multi-vesicular body vesicles, in which the protein FREE1 undergoes an intrinsically disordered domain (IDR)-dependent biomolecular condensate binds to membrane surface, driving membrane budding through capillary forces, following which, wetting-induced membrane asymmetries and line tension forces cause vesicle scission. Detailed modelling produce values for these effects that are reasonable and consistent with geometries and dimensions of the FREE1 condensates and vesicles *in vivo* and *in vitro* and of GUVs. Two general controls throughout the study are FREE1 with its IDR removed or replaced with the IDR of the known phase separating protein FUS. In all assays and imaging results, deletion of the FREE1 prevented but fusion to FUS IDR could restore FREE1 phase separation, vesicle formation *in vivo* and *in vitro*.

This is a very well written and well executed study of what appears a clear case of biomolecular condensate-driven morphogenesis of vesicles as has been shown now for a number of cases. I think it should be considered for publication in Nature.

A question I have concerns the very nice images of wortmannin-induced large MVBs on which we can see FREE1 associated with the membrane (Fig. 2a). My question is this: there don't appear to be any FREE1 generated vesicles inside the MVB. Is this just a result of the way the images were focused or does the increased diameter of the MVB result in a reduction in capillary forces that prevent membrane bending. On the same line, if it is the latter, can you estimate from your model the diameter of the FREE1 condensates, whether it could be visualized by a super-resolution microscopy method? If it were possible, it would be great as you could directly measure membrane bending geometry *in vivo* and determine if your results are consistent with theory.

Couple of minor things:

It would be useful to include some supplemental information about the IDR of FREE1 and why the authors chose the FUS IDR to substitute for it. Does it, like FUS, have a prion-like domain and if so, does it have a similar amino acid composition?

Among the examples of vesicle morphogenesis you should also cite the work on endocytic vesicles:

Bergeron-Sandoval, L. P. et al. Endocytic proteins with prion-like domains form viscoelastic condensates that enable membrane remodeling. *Proc Natl Acad Sci U S A* 118 (2021).
<https://doi.org:10.1073/pnas.2113789118>.

Referee #4 (Remarks to the Author):

The manuscript by Wang and co-authors reports that FREE1, a plant-specific ESCRT-I-related protein, is able to undergo liquid-phase separation, both under in vitro and in vivo conditions, and drive membrane deformation and scission in vitro. FREE1 is a FYVE-domain containing protein that plays important roles in both MVB trafficking and autophagy in Arabidopsis. It has been shown to interact with ESCRT-I components and be critical for the degradation of phytohormone receptors. Curiously, FREE1 is only found within a group of flowering plants, including Arabidopsis but not in other plants like grasses or non-flowering plants. The authors isolate phase-separation prone proteins by precipitation with biotinylated isoxazole and found enrichment in FREE1, VPS23, and VPS37. They also tested a total of 10 plant ESCRT proteins with predicted IDRs either fused to GFP (GFP-FREE1) or to mCherry and showed that only FREE forms condensates in vitro. The importance of the IDR in FREE1 function is supported by analysis in Arabidopsis. A FREE1 protein lacking the IDR is unable to rescue the mutant defects. The introduction of a different IDR unable to interact with ESCRT-I proteins into FREE1 is enough to restore function but not under stressful conditions. The authors also perform in vitro experiments to demonstrate that FREE1 can associate/wet GUV membranes and lead to their deformation and scission. A mathematical model also supports the notion that FREE1 should be able to induce vesiculation and membrane fission in MVBs.

I find this manuscript very interesting and relevant to understand previously underappreciated roles of phase-separating proteins in membrane deformation. This result would imply that phase-separating proteins could induce vesiculation of MVBs without the presence of cargo membrane proteins or even the late acting ESCRT proteins known to bend membranes in MVB sorting. I do have some concerns:

1) It is hard to understand how critical this process is for MVB sorting, considering that FREE1 is restricted to a small subgroup of plant species. How do the authors think that MVB membrane deformation occurs in other organisms, including plants with no FREE1, if FREE1 seems to be the only phase-separation prone protein within the ESCRTs? Also, it would be useful to include a discussion about previous studies showing that the presence of cargo is important for bending MVB membranes and what would be the role of other ESCRT proteins in MVB sorting if FREE1 can form ILV by itself. It is also unclear what is the role of FREE1 condensates in sequestering ESCRTs as highlighted by the authors if they are not required for MVB membrane deformation.

2) In general, the manuscript is missing some critical explanations on how experiments and simulations were performed to fully appreciate their implications. For example:

2a) In the analysis of proteins enriched in the precipitates with biotinylated isoxazole, the authors

checked for the enrichment of 8 ESCRT proteins (including FREE1) by western blotting. However, in the Mat & Methods section, they only mention antibodies against FREE1. Did they perform the precipitation in lines with tagged ESCRT proteins or they detected endogenous ESCRT proteins with specific antibodies not mentioned in the manuscript? This is important because, as the authors show, the presence of tags can alter the phase-separation properties of some proteins.

2b) The authors claim that FREE1 condensates can specifically sequester certain ESCRT subunits. Here again, the authors do not explain how these ESCRT proteins have been tagged or their expression level in plants.

2c) For the mathematical model, it would be very useful to clearly specify the sizes considered in the simulations. What sizes of ILVs are considered? Figure 3 mentions a radius for MVBs of 35nm but according to Extended Data Figure 7a, that is the full diameter of a typical ILV.

3) In a previous publication (Gao et al, 2014), it was shown that the FYVE domain of FREE1 was necessary for binding to membranes. The FYVE domain is located outside the IDR. In this manuscript, the authors show that a FREE1 version without its IDR but with its FYVE domain loses its capability to bind membranes. Can the authors reconcile these results? Related to this topic, the authors use wortmannin to increase the size of MVBs in Arabidopsis to better visualize the association between FREE1 and MVB membranes. However, wortmannin blocks the synthesis of PI3P and MBVs under these conditions should be largely devoid of PI3P. How do the authors explain that still FREE1 associate with MVBs? In other words, what would be providing the specificity of FREE1 to associate with endosomal and autophagosome membranes if PI3P is not required for its membrane association?

4) In line 231, the authors support their claim that FREE1 is internalized into ILVs by showing FREE1-positive puncta inside Con-A-treated vacuoles (Extended Data Fig 7d). At this resolution, individual ILVs should not be discernable. As shown before and mentioned in this manuscript, FREE1 is also transported to the vacuole for degradation by autophagy. The puncta shown in panel d are most likely autophagic bodies, not ILVs.

Author Rebuttals to Initial Comments:

We sincerely thank all four reviewers for taking the time to read and provide valuable
feedback on our manuscript. We were delighted to find that the comments of the reviewers
were generally positive and we greatly appreciate the suggestions to improve our analyses.
We have carefully read and considered all comments and have prepared a revised
manuscript to address these suggestions through an extensive set of additional experiments
and theoretical analyses. These include the generation of new transgenic plants, assessment
of embryo viability, gold particle-labelled electron microscopy analyses, EM and super
resolution imaging of newly-generated mammalian cell lines heterologously expressing
FREE1, updated 3D in silico simulations, lipidomic analyses and heterologous expression in
yeast cells. We feel that these have greatly improved the quality of our manuscript. Below,
we include our point-by-point responses to the reviewers' comments. Newly-added data that
have been integrated into the revised manuscript are also provided in this document for the
convenience of the reviewers, and we attached some additional results in support of our
responses to specific reviewer comments. We feel confident that these responses address
the reviewers' concerns.

Referee #1:

1. This paper presents evidence that a membrane-binding IDP (FREE1), a component of the
ESCRT-III machinery, is capable of driving internalization of vesicles into multi-vesicular
bodies in vivo, without the complete ESCRT-III machinery. Furthermore, they show that the
mechanism has a clear phenotype in vivo. The result is significant and of general interest
because of the high level of interest in protein condensates and to what degree they matter.
Here is a result where they appear to matter: the combination of their membrane-binding and
condensate-water tension is a concrete physical model of how they bind surfaces. Results:
FREE1 is enriched in isolates driven by b-isox. Of the enriched proteins, only FREE1 forms
condensates by itself. When the intrinsically disordered region (IDR) is removed, no
condensate is formed. The delta-IDR form is then used as a test of when FREE1-driven
condensation is operative in vivo. Delta-IDR does not rescue a previously known lethal
FREE1 knockout. An alternative (FUS) condensing IDR substitution of the FREE1 IDR
rescues knockout. An ESCRT-III protein (VPS23) does not interact with IDR/FUS-FREE1,
suggesting that the FUS-based rescue is independent of the entire ESCRT-III machinery.
The authors highlight the possibility that the entire ESCRT-III complex is not required for in
vivo scission.

**Response:** We thank the reviewer for their careful reading and positive assessment of
our manuscript.

2. It is challenging to show that any particular molecular mechanism is operative in vivo. For
example, the entire ESCRT-III machinery cannot be removed to show that MVBs can be
formed using FREE1 alone, or that scission is possible without ESCRT-III. The study here

relies on 1) reconstitution of internalization including scission in vivo, 2) the observation that
FUS-IDR recombinant proteins don't interact with an element of ESCRT-III but can rescue
FREE1 knockout, and 3) modeling.

I am convinced that condensation is a requirement for a critical stage of vesicle internalization
analogous to coat formation by clathrin or COPII. However, I don't see how this paper can
exclude that scission is still occurring in vivo by extended ESCRT machinery. The extended
ESCRT machinery likely has affinity for fission necks, where it performs its action and to
which its scaffolding shape is matched.

Response: We agree with the reviewer that it is challenging to exclude the possibility
that scission is still occurring in vivo by the extended ESCRT machinery. To tackle this
problem, we have performed a range of additional experiments.

1) The reviewer notes that it is not possible to remove the ESCRT machinery from cells
due to its essential nature. Indeed, plant embryos that are homozygous knock-outs for the
essential and conserved ESCRT-III subunit VPS2 (*vps2.1* mutants) are reported to not
develop beyond the early stages of embryogenesis (Katsiarimpa et al., Plant Cell 2011,
23:3026-40). We decided to take advantage of this lethality to test whether the ectopic
expression of condensing FREE1 is able to bypass the defective ESCRT machinery and
allow *vps2.1* mutants to progress to later developmental stages (Response Fig. 1e). While
~60% of *vps2.1* embryos arrested at the early heart and torpedo developmental stages
(Response Fig. 1a,b), those ectopically expressing IDR^{FUS}-FREE1 were significantly more
likely to progress to later stages of development and were approximately 50% larger than
those without FREE1 overexpression (Response Fig. 1b). Further, IDR^{FUS}-FREE1
expression largely rescued the ILV defect caused by VPS2 knock out (Response Fig. 1c),
and we detected IDR^{FUS}-FREE1 inside MVBs in these cells (Response Fig. 1d), which is
consistent with our model of FREE1-mediated scission. These results provide strong
evidence that FREE1 is able to compensate for major defects in ESCRT activity through its
role in ILV formation.

While FREE1 has no direct homologues in mammals (Gao et al., Curr Biol 2014,
24:2556-63), we observed that FREE1 forms condensates in COS-7 cells when expressed
(Response Fig. 2a). To test whether FREE1 can contribute to mammalian MVB formation,
we performed immuno-gold electron microscopy, which revealed that FREE1 localises to
MVB membranes (Response Fig. 2b). Using transmission electron microscopy, we observed
significantly more ILV-like structures inside MVBs upon expression of condensing FREE1
compared to non-condensing controls (Response Fig. 2c). Together with the fact that IDR^{FUS}-
FREE1 fully complemented the lethal phenotype of the *free1* knock-out mutant (Fig. 1f in the
previous submission) but did not interact with VPS23 (Fig. 2e-g in the previous submission),
these results further support our conclusion that FREE1 makes a significant contribution to
MVB biogenesis in the absence of the ESCRT-machinery. It is important to emphasize here
that all our observations show that only condensing FREE1 exerts this effect, which tallies

with our modelling showing that MVBs form ILVs when FREE1 condensates wet MVB
membranes.

Further, we employed super-resolution (structured illumination) microscopy to observe
that condensing FREE1 expressed in mammalian (COS-7) cells appears within MVBs
(Response Fig. 2d), which provides in vivo confirmation of our finding that FREE1
condensate-filled vesicles freely diffuse within GUVs in the absence of the ESCRT machinery
(Fig. 3f in the previous submission). Notably, we did not detect non-condensing FREE1^{ΔIDR}
in the MVB lumen in these experiments (Response Fig. 2d).

Together, these data provide several lines of additional evidence for condensate-
mediated ILV formation that we feel are very robust and strongly support the novel
condensate-dependent ILV formation mechanism proposed in our paper.

3. While FREE1 clearly has an impact on membrane localization of VPS23 (Fig 2d), it is not
clear that it is as decisive as the paper implies ("associates with SUVs only when FREE1 is
present") either in vitro or in vivo.

Response: In the revised manuscript, we now conclude that "we found that VPS23
association with SUVs largely increased when condensed FREE1 was present". To test
whether localization of VPS23 to MVBs in vivo depends on FREE1, we prepared protoplasts
from *FREE1/free1* and *IDR^{FUS}-FREE1/free1* plants and co-expressed VPS23 with the MVB
marker RHA1 (note that *IDR^{FUS}-FREE1* lacks the VPS 23 binding domain, which is encoded
within the FREE1 IDR). We observed that while VPS23 colocalized with RHA1-positive
structures in the *FREE1/free1* background, it was unable to localize to RHA1-positive MVBs
in the *IDR^{FUS}-FREE1/free1* background (Response Fig. 3). These data provide additional
evidence that condensed FREE1 is critical in determining the membrane localization of
VPS23 in living cells.

4. My expertise is modeling. The paper supplements the experiments with two models. One
model is a model with origins from the engineering field, with results shown in 3c and 3d. This
paper applies unique expertise and methodology to measure the time-dependence of
membrane reshaping (Fig 3d) incorporating surface tensions and viscosities. This is an
intriguing possibility but there is a major issue translating engineering problems to membrane
physics. From an engineering perspective, the membrane deformation is characterized by a
strain, with forces determined by a matrix of force constant terms that modulate pairs of
strains. Membrane biophysics has taken a somewhat different path to the Helfrich/Canham
Hamiltonian, mainly because the surface is a liquid crystal that cannot support a lateral shear
strain. Yet the neck at 30 milliseconds is a shape that I believe can be identified to be modeled
by a surface that penalizes lateral shear strain. (The authors write in the supplemental that a
shear modulus (K_s) is applied).

Response: We thank the referee for their appreciation of our dynamic model of
membrane shaping. With the aim to present a general model that is applicable to a diversity
of materials, we formally included the shear modulus for completeness. In agreement with
the comment of the referee, all data included in the manuscript were obtained by setting the
shear modulus $K_S = 0$ to account for the liquid nature of the membrane. The remaining area
modulus K_A effectively enforces the local inextensibility of the membrane. In the revised
manuscript, we explicitly clarify this detail. Our model thus accurately describes a fluid
membrane governed by a Helfrich Hamiltonian framework. The unusual shape of the neck
at 30 milliseconds in the former Fig. 3d was due to the fact that simulations were initially
performed in 2D (see our response to point 7 below). In the revised version, the simulations
were carried out in 3D (axisymmetric) and the resulting necks are small, as expected.

5. Instead, membrane necks are very small, where saddle curvature is penalized by a
Gaussian curvature modulus that cannot clearly be modeled by engineering moduli. Four
other minor comments on the model:

First, was the curved shape shown in Fig 3d chosen as the zero strain reference state of the
membrane, or is the zero strain shape flat? The zero strain reference state of a vesicle should
be flat, and this should result in a sphere as the minimum energy state, not something
oblong.

Response: The simulations describe a fluid membrane so there is no reference state.
We agree with the referee that the minimum energy state of a vesicle would be a sphere if
the vesicle volume is not conserved and the vesicle is therefore fully inflated. In the original
manuscript, we choose not a shape of minimum energy but rather a specific degree of vesicle
deflation (in other words, excess membrane area), which was set in a way that the engulfment
of the condensate by the membrane is not limited by an area constraint.

In the revised manuscript we have addressed the reviewer's concerns by choosing initial
GUV and MVB shapes that minimize the bending energy (under the constraint of area and
volume conservation, Response Fig. 4a). We obtained these shapes by running a
condensate-free simulation up to the stationary state. Consistent with published data we find
that non-spherical, oblate shapes are the states of minimum energy for slightly deflated
vesicles (e.g. Fig. 9 in Seifert et al., 1991 Physical review A, 44, 1182).

6. Second, it appears a uniform viscosity was chosen. Yet water has a viscosity one thousand
150 times smaller than a condensate. How would a correct specification of water viscosity affect
kinetics?

Response: A uniform viscosity was chosen for two key reasons: in the interests of
simplicity, as suggested by the reviewer, and also as lower ambient viscosities should have
a minor effect. In the revised version we now explicitly include results illustrating the changes
in invagination dynamics with reduced ambient viscosities by a factor of 10-1000 (Response

Fig. 4b,c) that account for reduced viscosity of the surrounding solution compared to
condensates (in cells, this is the cytosol; in GUVs, this is a water-like buffer). During these
simulations we noticed that the shape of the ILV invagination intermediates changes
marginally by reducing viscosities (Response Fig. 4d). We additionally include shapes of
invagination intermediates for both GUV and MVB membranes in the revised manuscript
(Response Fig. 4b,d). We thank the referee for this valuable suggestion which has allowed
162 us to refine our model.

7. Third, the authors must be clearer about the dimensionality of the system. It is very hard
to tell if this work uses three dimensional modeling, if it uses two dimensional modeling using
axial symmetry, or if it uses purely 2 dimensional modeling with the system isotropic "out of
the page".

Response: We thank the referee for pointing out this lack of clarity on our part. In the
original manuscript, all simulations were 2D as this approach allowed us for the first time to
model wetting of multiple condensates on a single GUV membrane. We note that, despite
being 2D, these models closely reproduced wetting phenomena observed experimentally
(except the shape of the neck; see response to point 4 above). In the revised manuscript, we
have adopted an enhanced axisymmetric 3D system (Response Fig. 4a): in all but one figure
in the revised manuscript, we exclusively use this 3D system for modelling wetting of single
condensates. The 2D system is now only used in Extended Data Fig. 8b of the revised
manuscript, where we modelled the interaction of multiple condensates with the GUV. We
have updated the text to ensure that system dimensionality is clearly indicated.

8. Fourth, what are the flows in Fig 3c. Is something flowing through the bilayer? If so, this
shouldn't be. If not, to me it's very confusing.

Response: We thank the referee for this comment. There is nothing flowing through the
bilayer. Streamlines indicate movement of modelled system components and illustrate that
the membrane moves consistently with fluid flow. We acknowledge that this type of
visualization can be confusing. Based on the reviewer's feedback, we have updated the figure
caption to improve our communication of the results. The original Fig. 3c is now included as
Extended Data Figure 8c.

9. While I am very much excited by the model, I did not see its important connection to the
main result of the paper, especially including my confusion about the points above.

Response: We thank the referee sharing our excitement and suggesting that we need to
better explain the in silico approach in the context of our experimental results. As we prepared
the revised manuscript, we had initially intended to record spontaneous condensate wetting
events in vitro and to correlate experimentally determined dynamics with our computational
model, in line with the suggestions of the reviewer. However, these experiments turned out

to be more challenging than anticipated. Most notably, FREE1 forms membrane bound
condensates surprisingly quickly when mixed with GUVs; membrane binding was near-
immediate and much faster than other protein condensates that we have previously assessed.
We found that GUVs were associated with multiple wetting condensates even at the point
that they settled at the bottom of the observation chamber. As it is extremely difficult to image
GUVs before they have settled, this makes it practically impossible to record spontaneous
wetting events by monitoring GUVs in a conventional formation chamber setup. As an
alternative approach, we sought to guide condensates towards GUVs using optical trapping.
While non-trapped or freshly trapped FREE1 condensates are liquid-like (as indicated by
coalescence dynamics, Fig. 1), we found that stably optically trapped condensates are unable
to coalesce or wet GUV membranes during a relevant time frame (seconds). This effect also
prevented us from using active rheology to describe FREE1 condensate physical properties.
Further, the size of ILVs is well below optical resolution, preventing assessment of their
formation dynamics by live cell imaging of MVBs. As a result, modelling is the only available
strategy to understand wetting dynamics and budding intermediate morphologies for both the
GUVs and MVBs. Being able to retrieve condensate-membrane intermediate shapes and
their dynamics goes well beyond conventional equilibrium shapes and makes our
computational model an important addition to the manuscript, as well as a valuable resource
to colleagues in the field. We have highlighted these points in the revised manuscript and
trust that the reviewer will appreciate the experimental constraints of this system, as well as
the value of modelling in tackling such challenging questions.

10. The second model is a traditional membrane biophysics model applying standard
modeling constants based on theory from the Lipowsky group. I strongly warn the authors
against over-interpreting the energetics of leaflets using continuum theory near the pinching
point (Supplemental Eqs. 5 and 6). There are three related issues: One, for a very small exo-
or endo-cytic pore, while the mean curvature of the leaflets might be zero at the bilayer
midplane, the curvature of the inner (collapsing) leaflet diverges. This difference between
leaflet curvature and bilayer curvature only applies at very high curvatures (exactly this
scenario). From Agudo-Canalejo and Lipowsky, 2016, the source of the equations from which
the authors derive subsequent results: "Predicting whether the closed neck will break, leading
to fission, or whether it will remain intact, leaving the topology of the membrane unchanged,
lies beyond the realm of applicability for the continuum membrane model used here. As an
example, our continuum model allows for the formation of infinitely narrow membrane necks,
whereas the diameter of a real neck must exceed twice the membrane thickness,
[approximately] 4–5 nm."

Two, for topological changes of the membrane the sum Gaussian curvature changes
discontinuously, and we currently don't know where this model energy comes from or goes

(see Siegel and Kozlov 2004 Biophys J 87 366 Fig. 2 for an illustration and the Appendix for
how to analyze the leaflets).

Three, proteins simply may not fit around a very small pore, especially in the endocytic case.
Yet their effect at this critical site appears to still be included in the theory.

The sum of these observations is that we really don't know what's going on for very small
pores. We do know that the cell typically brings in extremely powerful actors like dynamin on
the outside to complete fission. Not only is the scission theory likely invalid (engulfment is
fine), but scission may still be a transition state with a significant activation energy and still
be consistent with their experiments. I believe the process is effectively irreversible after the
final interior-coated particle is made. The authors' proof-of-principle theory would be far more
likely to stand the test of time if, instead of a spontaneity relation for full scission, the authors
used their theory to place the scission-inducing line tension in the context of the line tension
required to stabilize incomplete internalization. I think this may be done with minor shifts in
language in the main text and severe caveats placed on the applicability of elasticity theory
to fission.

**Response:** We thank the referee for carefully considering the details of our model. We
are aware of the power and the limitations of the Helfrich continuum model. Indeed, Agudo-
Canalejo, who wrote the passage quoted from Agudo-Canalejo & Lipowsky 2016, is one of
the co-authors of this study. In our approach, we circumvent the very real challenges
associated with a microscopic theory of membrane scission discussed by the reviewer.
Instead, we calculate what is the neck constriction force within Helfrich theory, and make the
(what we feel is very reasonable) assumption that scission should happen spontaneously
when this force reaches some critical value. The value of this critical scission constriction
force (f_*) has previously been quantified experimentally as $f_* \approx 25$ pN (Steinkühler et al.
Nature Comm 2020, 11:905.). Importantly, Steinkühler et al. verified that the neck constriction
force f is indeed given by the expression derived from Helfrich theory. By calculating f for
condensate-induced membrane necks, we obtain the line tension required to stabilize
incomplete internalization (i.e. invagination without scission) and the line tension to scission
necks. These line tensions are shown as grey and blue lines in all revised stability diagrams.
We have modified the text accordingly and also now include stability diagrams for a range of
physiologically relevant MVB and ILV sizes as well as wetting-induced membrane
asymmetries (Response Fig. 5b-e).

In the revised version, simulations with non-zero line tension have also been performed.
Given that in the simulations the droplet-cytosol interface is diffuse rather than sharp and full
closure of the neck is impossible due to discretization issues, a perfect match between
simulations and the theory cannot be expected. However, consistent with the theoretical
prediction, the in silico neck radius is observed to drop from ~ 6.5 nm for $\lambda=0$ to ~ 1.5 nm for
$\lambda=10$ pN (Response Figs. 4f and 5e).

To respond to specific issues raised in more detail:

- • We agree that, as the neck radius approaches zero, adjustments need to be made
to the model arising from higher order curvature terms (or equivalently, the
thickness of the membrane). We note that the passage quoted by the referee goes
on to say that “corrections to the model will stabilize the neck size at a finite non-
zero value”. In mathematical terms, the constriction force f that we calculate from
the Helfrich model is what governs the linear behaviour in an expansion of $E(R_{ne})$
in powers of R_{ne} , i.e. $E(R_{ne}) \approx fR_{ne}$. The higher order corrections should lead to
terms manifesting as negative powers of R_{ne} , which stabilize neck size (Response
Fig. 5a). Importantly, the constriction force calculated from the Helfrich model still
“pushes” against these higher order terms and squeezes the neck. The more
sophisticated models mentioned by the reviewer, accounting for the bilayer nature
of the membrane, the transition states for fission, etc., could potentially provide a
microscopic derivation of the critical force f_* , but we do not attempt this and simply
use the experimentally measured value of f_* .
- • As the reviewer notes, conventional fission models propose that filamentous
protein machineries assemble around small pores. However, our proposed
mechanism does not require neck fitting scission machineries because wetting
condensates constrict the neck indirectly from the inside of the membrane
invagination by generating a line tension (described by the three-phase contact
line) and changing the membrane asymmetry locally. In the revised manuscript,
we highlight this important point and also discuss that for several important cellular
processes (including the formation of autophagosomes, protein storage vacuoles
and secretory granules), condensates in fact have physiological roles while
essential scission machineries have not yet been exclusively implicated in the
scission event. In the revised manuscript, we calculated as a reference the line
tensions required for scission for both budding directions (inward and outwards),
for three different condensate sizes (30 nm, 300 nm and 3000 nm) and a range of
relative membrane sizes (Response Fig. 5f-h). We note that the condensate-
mediated fission mechanism could also have implications in the context of
compartmentalization dynamics in the origin of life, which we also now mention in
the manuscript.

11. I am unsure if readers will understand the connection to spontaneous curvature. I think
the model would be just as strong focusing clearly on surface tensions giving rise to line
tension and mentioning that protein-lipid associations change spontaneous curvature, likely
supporting fusion and lowering the requirement of a strong line tension.

Response: We thank the reviewer for this excellent suggestion. We believe that this
simplification will help the majority of readers to appreciate our model, and have modified the

text accordingly. For expert readers, we have retained a complete description of theory,
including spontaneous curvature details, in the SI.

12. The narrative of the article was very well put together, with the authors frequently
anticipating my next thought immediately with a well described experiment. In my view I would
advise strongly against any revision of the narrative structure.

**Response:** We appreciate the reviewer's positive assessment of our manuscript's
structure.

13. In summary, I am very excited about this paper and found the set of experiments to be
convincing about the necessity of phase separation but only suggestive about the simplicity
of the scission mechanism, based both on the nearly insurmountable ambiguity of in vivo
experiments and the failure of membrane continuum theory to address scission. The
experiments have not dramatically shifted my thinking about the necessity of complicated
scission machinery in vivo, but they have certainly opened the possibilities. It has continued
to shift my thinking about the necessity of complicated coats (like clathrin). Put in the context
of what I believe is a very exciting and challenging frontier of experiment and theory, this
paper is a critical contribution. In many places, the authors write carefully to indicate that the
condensate-driven scission mechanism is only suggested.

**Response:** We thank the reviewer for pointing out the nearly insurmountable ambiguity
of in vivo experiments and their detailed, very constructive feedback. In our revision, we now
provide multiple additional lines of evidence, which we are confident lend further support for
the condensate-mediated scission mechanism proposed in this study.

14. The two places where I disagreed with language were the applicability of continuum
theory to scission, and the exclusion of the full ESCRT-III machinery from scission by the
weak association of VPS23 and FUS-IDR modified FREE1 as proof that VPS23-associated
elements would not participate in scission.

**Response:** We have rephrased the text in both places. Thank you for helping to improve
the clarity of our work.

**Referee #2 (Remarks to the Author):**

1. The major experimental result of the study is that the plant ESCRT component, FREE1,
forms liquid condensate. The FREE1 condensate droplets binds and can be engulfed by lipid
bilayers of GUV. Moreover a membrane neck connecting an engulfed droplet with the initial
GUV membrane can undergo fission. In addition, FREE1 condensation was observed also
in cells. The major message of the study is that droplets of FREE1 condensate droplets can
mediate membrane budding and fission in the process of formation of internal vesicles of

MVBs. In addition, theoretical modeling is presented with a goal to support the observations.
I have the following reservations concerning this work.

Response: We appreciate the reviewer's careful reading of the manuscript and detailed
feedback. Based on the reviewer's comments, we have designed and conducted new
experiments, as described below, and hope that the new data we provide addresses the
reviewer's concerns.

2. The major proposal concerning the intracellular membrane budding/fission by FREE1
liquid droplets remains a speculation. The authors should demonstrate existence within cells
(MVBs) of FREE1 droplets covered by membranes. Demonstration of such covered droplets
in GUVs is not sufficient.

Response: We thank the reviewer for this suggestion. To determine if FREE1
condensates exist within MVBs, we first conducted further analyses of wortmannin-enlarged
MVBs. In these experiments, we were able to observe FREE1 condensates within MVBs
(Response Fig. 6a). This finding is consistent with a previous study that reported that FREE1
puncta entered the lumen of MVBs (Xia et al., Plant Cell 2020, 32:3290-3310).

Second, we co-expressed Rab5^{Q79L}, an MVB-localized protein that was previously
reported to enlarge MVBs (Trajkovic et al., Science 2008, 319:1244-1247), with GFP-FREE1
or GFP-FREE1^{ΔIDR} in mammalian (COS-7) cells. Using super-resolution (structured
illumination) microscopy, we observed FREE1 condensates inside MVBs. In contrast, GFP-
FREE1^{ΔIDR} signal was excluded from MVB lumen (Response Fig. 2d).

Third, we tested whether the ectopic expression of condensing FREE1 is able to bypass
the lack of function of a defective ESCRT machinery and allow *vps2.1* mutants to progress
to later developmental stages (Response Fig. 1e). While ~60% of *vps2.1* embryos arrested
at the early heart and torpedo developmental stages (Response Fig. 1a,b), those ectopically
expressing IDR^{FUS}-FREE1 were significantly more likely to progress to later stages of
development and were approximately 50% larger than those without FREE1 overexpression
(Response Fig. 1b). Further, IDR^{FUS}-FREE1 expression largely rescued the ILV defect
caused by VPS2 knock out (Response Fig. 1c), and we detected IDR^{FUS}-FREE1 inside MVBs
in these cells (Response Fig. 1d), which is consistent with our model of FREE1-mediated
scission. These results provide strong evidence that FREE1 is able to compensate for a major
defect in ESCRT activity through its role in ILV formation.

3. The whole theoretical part of the article does not advance the understanding. It is not new
in its essence on one hand, and does not provide any new insight into the system, on the
other. In addition, it is confusing. Specifically, -lines 183-186. This criterion of a particle
engulfment by a membrane has been derived in multiple works and does not need any
complex consideration of the system dynamics. In general the simulations of the dynamics
(Fig.3C) are not justified by the experimental results and are, therefore, unnecessary here.

The purpose of the computations of the shapes (Fig.3d) is unclear. Such calculation were
done in multiple previous works so that the present one does not provide any new
understanding.

Response: We thank the reviewer for this feedback but must respectfully disagree. The
only theoretical predictions available in the literature are stationary shapes of simplified
scenarios based on energy minimization. None of these approaches resolve the dynamics of
the engulfment process. Therefore, the engulfment timescales and intermediate shapes
uncovered in our paper are novel and will be of significant interest to researchers in fields
ranging from capillarity and membrane shaping to intracellular degradation processes. We
note that reviewer 1 explicitly states that “this paper applies unique expertise and
methodology to measure the time-dependence of membrane reshaping” and cordially invite
the reviewer 2 to cite specific examples of theoretical work that pre-date our work that we
may have overlooked.

We believe that the relative lack of progress in this area reflects the technical difficulties
in studying wetting interactions on this scale: engulfment dynamics and morphologies cannot
be determined empirically as ILVs of MVBs are below optical resolution in vivo, and ILVs in
GUVs form very rapidly, presenting serious technical challenges in the visualisation of the
initial interactions between condensate and membrane. We adopted our innovative
computational model to address these limitations and feel that it is a powerful technique that
provides a clear account of the principles underlying the phenomena we observe both in vitro
and in vivo. In addition, our developed model describes the free surface of the condensate
by a diffuse interface. This is in contrast to previous theoretical approaches, which are all
based on sharp interface descriptions. Therefore, we resolve the diffusive nature of the
condensate interface for the first time in the context of membrane wetting. In the revised
manuscript, we have rewritten the introduction to our model and explained the experimental
limitations in more detail to better communicate the necessity and advantages of simulations
to understand condensate-mediated scission. We have also included simulation movies
(Supplementary Videos 1-4) as supporting information to allow readers to better understand
this powerful tool.

4. The fission phase diagram is very speculative. The used values of the model parameters
are not well justified. In particular, (i) the values of the tensions are estimated from the
relationship (Eq.8) of the Supplement, which implies the membrane under the droplet to be
flat, which is not the case. (ii) The origin of the line tension is obscure. I cannot think of any
reasonable microscopic origin of the line tension in this particular system. A reference to a
general property of a three-phase boundary is insufficient to justify the existence of the line
tension not to mention an estimation of its value. (iii) The criterion of membrane fission used
here is also rather unconvincing. The authors use the previously published relationships (Eqs.
1-6) of the second part of the Supplement, which were derived for tension-free membranes,

while in the current work, the membranes (at least one of them if not both of them) are under
considerable tension. It is easy to see that the presence of tension qualitatively changes the
expression of the neck expanding force.

Response: We thank the reviewer for sharing their concerns. To respond to each point:

(i) The theory developed in this study is general and can account for scenarios in which
the radius of the condensate droplet R_d is comparable to the radius of the vesicle R_v ,
meaning that the contacted membrane need not be flat (indeed, we designed this model
based on the assumption that the initial membrane is not flat). In the new Fig. 4a and
Extended Data Fig. 10a of the main text (also included as Response Fig. 5b,c here), we now
include the radius of the MVB membrane as a parameter in the scission diagram.

(ii) From a physical perspective, molecules along the contact line are exposed to a
different environment than those far away from the contact line. Therefore, a non-zero line
tension can be expected in any wetting process. This contribution to the overall free energy
of the system scales with the length of the contact line. While line tensions have a negligible
effect on the wetting of large droplets (on the micrometre scale), they can have a significant
effect on the wetting of smaller (nanometre-scale) droplets (Ghosh et al., Nature Comms
2023, 14:615). This is demonstrated by the fact that the line tension energy scales as λR_d ,
while the surface tension energy scales as $\sigma_{\beta\gamma} R_d^2$. By comparing these two contributions, we
see that the line tension contribution starts to be significant and ultimately dominate for
sufficiently small droplets with $R_d \sim \lambda / \sigma_{\beta\gamma}$ or smaller. In any case, it is not our aim to precisely
determine the line tension λ ; rather, we note that the values of line tension that can result in
neck scission are well within realistic orders of magnitude for the line tension of three-phase
contact lines and consistent with our ILV formation experiments

Besides the review by Law et al. already included in the original version, we now also
cite Ghosh et al. (Nature Comms 2023, 14:615). In this work, dissipative particle dynamics
(DPD) simulations of the engulfment of nanometre-scale droplets by lipid vesicles measured
line tensions at the three-phase contact line which, depending on the composition of the
membrane, could have positive and negative values ranging from -130 pN to +100 pN.

(iii) All previously published equations for the forces at the neck were devised in studies
of uniform membranes or uniform membranes enwrapping solid particles. In all of these
cases, the resulting equations are independent of membrane tension (Jülicher & Lipowsky,
Phys Rev E 1996, 53:2670-2683; Agudo-Canalejo & Lipowsky, Soft Matter 2016, 12:8155-
8166; Lipowsky, J Phys Chem B 2018, 122:3572-3586). Our manuscript develops equations
that, for the first time, describe membrane wetting condensates and wetting-mediated
membrane inhomogeneities as drivers for membrane scission. We do not assume tension
free membranes: as in previous studies, we find that membrane tension simply does not play
a role in the stability of the neck.

For these reasons, we feel that our model is not speculative, and is based on reasonable
and physiologically-relevant parameters that point to the likelihood of condensate-mediated
scission occurring in vivo. Moreover, we note that the theory of neck scission by constriction
forces has already been experimentally validated for the particular case of uniform
membranes in Steinkühler et al. (Nature Comms 2020, 11:905.)

**Referee #3:**

1. Feng et al. present compelling in vivo and in vitro data to support a model for ESCRT-and
ATP-independent formation of multi-vesicular body vesicles, in which the protein FREE1
undergoes an intrinsically disordered domain (IDR)-dependent biomolecular condensate
binds to membrane surface, driving membrane budding through capillary forces, following
which, wetting-induced membrane asymmetries and line tension forces cause vesicle
scission. Detailed modelling produce values for these effects that are reasonable and
consistent with geometries and dimensions of the FREE1 condensates and vesicles in vivo
and in vitro and of GUVs. Two general controls throughout the study are FREE1 with its IDR
removed or replaced with the IDR of the known phase separating protein FUS. In all assays
and imaging results, deletion of the FREE1 prevented but fusion to FUS IDR could restore
FREE1 phase separation, vesicle formation in vivo and in vitro. This is a very well written and
well executed study of what appears a clear case of biomolecular condensate-driven
morphogenesis of vesicles as has been shown now for a number of cases. I think it should
be considered for publication in Nature.

**Response:** We thank reviewer #3 for the positive feedback on our work.

2. A question I have concerns the very nice images of wortmannin-induced large MVBs on
which we can see FREE1 associated with the membrane (Fig. 2a). My question is this: there
don't appear to be any FREE1 generated vesicles inside the MVB. Is this just a result of the
way the images were focused or does the increased diameter of the MVB result in a reduction
in capillary forces that prevent membrane bending.

**Response:** To address this concern, we acquired z-stack images of wortmannin-induced
large MVBs and observed FREE1 condensates within MVBs (Response Fig. 6a). Further, we
employed immuno-gold labelling to detect FREE1 inside the MVB lumen by electron
microscopy (Response Fig. 1d). These findings are consistent with a previous study which
found that FREE1 entered the lumen of MVBs (Xia et al. Plant Cell 2020, 32:3290-3310).

Moreover, we co-expressed human Rab5^{Q79L}, an MVB-localized protein that was
reported to enlarge MVBs (Trajkovic et al., Science 2008, 319:1244-1247), with GFP-FREE1
or GFP-FREE1^{ΔIDR} in COS-7 cells. Using super-resolution microscopy, we were able to
clearly see that FREE1 condensates appeared inside enlarged MVBs (Response Fig. 2d). In
contrast, GFP-FREE1^{ΔIDR} signal was excluded from the MVB lumen (Response Fig. 2d).

Increased MVB size has two effects on ILV formation: first, assuming no change in the
MVB reduced volume (or MVB relative excess area), a larger MVB will have more available
area to engulf condensates if the condensate size is kept constant. Second, equation 8 in the
Supplementary Methods shows that the force at the neck includes a contribution that is
expressed as $1/R_v$, where R_v is the radius of the MVB. Thus, if everything else is kept
constant, the larger the MVB, the smaller is the force exerted at the neck, which makes
scission more difficult. We now account for this MVB size effect in the updated Fig. 4a and
Extended Data Fig. 10a of the main text (also included as Response Fig. 5b,c here) and
thank the reviewer for pointing out the relevance of considering MVB size effects.

2. On the same line, if it is the latter, can you estimate from your model the diameter of the
FREE1 condensates, whether it could be visualized by a super-resolution microscopy
method? If it were possible, it would be great as you could directly measure membrane
bending geometry in vivo and determine if your results are consistent with theory.

Response: We thank the referee for this suggestion. The current optical resolution of
light microscopy is unfortunately too low to allow a direct comparison of bending geometries
in vivo and in silico. As detailed in the previous point, we observed FREE1 condensates
inside MVBs in both plants and other systems using super-resolution microscopy and
immuno-EM (Response Figs. 1d, 2d, 6a). Further, we quantified the size of ILVs in MVB
using transmission electron microscopy (Extended Data Fig. 9c) and used the measured ILV
diameter of 35 nm as a parameter in our computer simulations and theoretical model. In the
revised version, we explain our use of empirical data to inform and refine our modelling and
simulation approaches in more detail.

3. Couple of minor things: It would be useful to include some supplemental information about
the IDR of FREE1 and why the authors chose the FUS IDR to substitute for it. Does it, like
FUS, have a prion-like domain and if so, does it have a similar amino acid composition?

Response: We thank the reviewer for these suggestions. Indeed, we chose the FUS IDR
because it shares a similar amino acid composition with the FREE1 IDR. Additionally, both
are prion-like and similar in length (Response Fig. 7a,b). For the manuscript revision, we
generated an additional FUS variant (IDR^{FUSm}-FREE1) by substituting tyrosine residues
within IDR^{FUS} with serine (Wang et al., Cell 2018, 174: 688–699) to abolish the phase
separation capacity of IDR^{FUS}. Compared to IDR^{FUS}-FREE1, IDR^{FUSm}-FREE1 lost phase
separation in vitro (Response Fig. 7c). This variant also failed to rescue *free1* mutant lethality
of plant embryos (Response Fig. 7d), supporting our finding that the phase separation of
FREE1 is critical.

We also generated a new FREE1 chimeric protein by replacing the FREE1 IDR with that
of FLOE1 (floating ice 1), a prototypical phase separating protein that requires this IDR to
form condensates (Dorone et al., Cell 2021, 184:4284-4298). IDR^{FLOE1} bears no apparent

similarity to IDR^{FREE1} or IDR^{FUS} aside from the presence of prion-like domains, a similar length
and the overall amino acid composition (Response Fig. 7a,b). As for wild-type FREE1,
chimeric IDR^{FLOE1}-FREE1 phase separated *in vitro* (Response Fig. 7c). Importantly,
IDR^{FLOE1}-FREE1 rescued FREE1 knock-out (*free1*) seedling lethality (Response Fig. 7d).
Together, these additional data further support for our conclusion that the ability of FREE1 to
undergo condensation rather than IDR sequence is the critical feature bestowing FREE1
function. As suggested, we have included these additional data in the supplemental
information of the revised manuscript.

4. Among the examples of vesicle morphogenesis you should also cite the work on endocytic
vesicles: Bergeron-Sandoval, L. P. et al. Endocytic proteins with prion-like domains form
viscoelastic condensates that enable membrane remodeling. Proc Natl Acad Sci U S A 118
(2021).

Response: We thank the reviewer for pointing out that we missed citing this excellent
work. We have added this citation in the revised text.

**Referee #4** (Remarks to the Author):

1. The manuscript by Wang and co-authors reports that FREE1, a plant-specific ESCRT-I-
related protein, is able to undergo liquid-phase separation, both under *in vitro* and *in vivo*
conditions, and drive membrane deformation and scission *in vitro*. FREE1 is a FYVE-domain
containing protein that plays important roles in both MVB trafficking and autophagy in
Arabidopsis. It has been shown to interact with ESCRT-I components and be critical for the
degradation of phytohormone receptors. Curiously, FREE1 is only found within a group of
flowering plants, including Arabidopsis but not in other plants like grasses or non-flowering
plants. The authors isolate phase-separation prone proteins by precipitation with biotinylated
isoxazole and found enrichment in FREE1, VPS23, and VPS37. They also tested a total of
10 plant ESCRT proteins with predicted IDRs either fused to GFP (GFP-FREE1) or to
mCherry and showed that only FREE forms condensates *in vitro*. The importance of the IDR
in FREE1 function is supported by analysis in Arabidopsis. A FREE1 protein lacking the IDR
is unable to rescue the mutant defects. The introduction of a different IDR unable to interact
with ESCRT-I proteins into FREE1 is enough to restore function but not under stressful
conditions. The authors also perform *in vitro* experiments to demonstrate that FREE1 can
associate/wet GUV membranes and lead to their deformation and scission. A mathematical
model also supports the notion that FREE1 should be able to induce vesiculation and
membrane fission in MVBs. I find this manuscript very interesting and relevant to understand
previously underappreciated roles of phase-separating proteins in membrane deformation.
This result would imply that phase-separating proteins could induce vesiculation of MVBs
without the presence of cargo membrane proteins or even the late acting ESCRT proteins
known to bend membranes in MVB sorting.

Response: We appreciate the careful reading and thorough review of our manuscript by
the reviewer. All comments provided by this reviewer, which are addressed below, are very
valuable and we feel have led to a marked improvement in the quality of our manuscript.

2. I do have some concerns: 1) It is hard to understand how critical this process is for MVB
sorting, considering that FREE1 is restricted to a small subgroup of plant species. How do
the authors think that MVB membrane deformation occurs in other organisms, including
plants with no FREE1, if FREE1 seems to be the only phase-separation prone protein within
the ESCRTs?

Response: The reviewer raises an important point that we did not address sufficiently in
our initial submission. We believe that the previous report stating that FREE1 is conserved
only amongst eudicots (Gao et al., Curr Biol 2014, 24:2556-63) is not accurate: our BLAST
analyses revealed FREE1 homologues in all representative land plants, but not in algae
(Response Fig. 8a). These homologues exhibited a very similar domain architecture: a FYVE
domain, a C-terminal coiled coil (CC) domain, and an N-terminal IDR (Response Fig. 8b).
While the FYVE domain and CC were highly conserved, the IDR sequences are highly
variable (Response Fig. 8c), likely explaining why FREE1 homologs have not yet been
identified outside eudicots (Gao et al., Curr Biol 2014, 24:2556-63). This is consistent with
our finding that FREE1 retains function when the original IDR^{FREE1} is replaced with unrelated
but condensate-forming IDRs from FUS or FLOE1 (IDR^{FUS} and IDR^{FLOE1}; Fig. 1f and
Response Fig. 7d). We were also able to highly enrich FREE1 homologs of other species
using b-isox (Response Fig. 8d). Together, these results suggest that FREE1 homologs are
found in many plant species and that FREE1 condensate-mediated membrane scission is
likely a conserved mechanism in land plants.

The physical nature of this mechanism also likely mediates membrane scission in other
ESCRT-dependent cellular processes unrelated to FREE1. For example, in budding yeast,
the ESCRT-0 proteins Vps27 and Hse1 form condensates in the presence of polyubiquitin.
These condensates localize to the vacuole membrane and can facilitate the direct
internalization and degradation of cargoes within vacuoles (Banjade et al., Sci Adv 2022,
8:eabm5149) in a process that is topologically equivalent to ILV formation of MVBs. Another
example is the mammalian ESCRT protein ALIX, which undergoes condensation *in vitro* and
*in vivo* and, similarly to FREE1, partitions other ESCRT proteins (Elias et al., Sci Adv 2023,
9: eadg3913). While the membrane remodelling capability of ALIX condensates has not yet
been determined, it is plausible that these ESCRT condensates mediate membrane
invagination and scission in a manner similar to FREE1.

Finally, condensate-mediated membrane scission may operate in cellular processes
involving condensates in a range of organisms. Examples include the scission of condensate-
enclosing autophagosomes (Agudo-Canalejo et al., Nature 2021, 591:142-146), the scission
of condensate-containing secretory granules from the trans-Golgi network (Parchure et al.,

JCB 2022, 221:e202206132) and tonoplast scission by vacuolar storage protein condensates
during plant embryogenesis (Kusumaatmaja et al., PNAS 2021, 118: e2024109118).
Because endocytic machineries also form membrane-bound condensates (Dragwidge et al.,
Nat Cell Biol 2024, s41556-024-01354-6; Bergeron-Sandoval et al., PNAS 2021, 118:
e2113789118), it is tempting to hypothesize that these condensate-membrane interactions
contribute force to scission membranes during endocytosis. We have addressed these
interesting points in the discussion of the revised manuscript. Further, as a reference we
calculated the line tensions required for scission for a range of physiological relevant
scenarios including both budding directions (inward and outwards), three different
physiological relevant condensate sizes (30 nm, 300 nm and 3000 nm) and a range of relative
membrane sizes (Response Fig. 5f-h).

2. Also, it would be useful to include a discussion about previous studies showing that the
presence of cargo is important for bending MVB membranes and what would be the role of
other ESCRT proteins in MVB sorting if FREE1 can form ILV by itself. It is also unclear what
is the role of FREE1 condensates in sequestering ESCRTs as highlighted by the authors if
they are not required for MVB membrane deformation.

Response: In the wild type background, it is plausible that condensate-dependent and
ESCRT-dependent pathways are interconnected and that membrane scission can be
achieved through contributions from both pathways. It has been shown that the high local
concentration of actin monomers present in condensates, which arises due to partitioning,
enhances the formation of actin filaments in condensates (Graham et al., Nat Physics 2023,
19:574–585). Similarly, by recruiting ESCRT machinery components via partitioning (Fig. 2,
Extended Data Figure 6), FREE1 condensates might enhance the ESCRT-dependent
pathway. Further, it is plausible that FREE1 condensates can sequester ESCRT components
for degradation. This hypothesis is supported by the finding that the MVB pathways
contributes to the degradation of the ESCRT-I subunit VPS23 (Xia et al., Plant Cell 2020,
32:3290-3310). We thank the reviewer for this comment and include a discussion of this point
in the revised manuscript.

3. In general, the manuscript is missing some critical explanations on how experiments and
simulations were performed to fully appreciate their implications. For example: a) In the
analysis of proteins enriched in the precipitates with biotinylated isoxazole, the authors
checked for the enrichment of 8 ESCRT proteins (including FREE1) by western blotting.
However, in the Mat & Methods section, they only mention antibodies against FREE1. Did
they perform the precipitation in lines with tagged ESCRT proteins or they detected
endogenous ESCRT proteins with specific antibodies not mentioned in the manuscript? This
is important because, as the authors show, the presence of tags can alter the phase-
separation properties of some proteins. b) The authors claim that FREE1 condensates can

specifically sequester certain ESCRT subunits. Here again, the authors do not explain how
these ESCRT proteins have been tagged or their expression level in plants.

Response: We apologise for the lack of detail regarding these experiments. To clarify,
we performed two different b-isox precipitations. First, we used ten-day-old *A. thaliana* Col-0
seedlings and the FREE1 antibody for western blotting. We also tagged each ESCRT protein
individually with a widely-used small epitope, FLAG, as antibodies are not available for plant
ESCRTs, expressed these proteins transiently in *Nicotiana bethamiana* leaves, and
performed b-isox precipitation. Western blotting was performed using the anti-FLAG antibody.

For in vitro partitioning, VPS23, VPS28 and VPS37 were tagged with mCherry at their
C-termini; these proteins are known to be functional in vivo when tagged in this manner (Yu
et al., Mol Plant 2016, 9:1570-1582; Liu et al., Plant J 2020, 104:1617-1634; Spallek et al.,
PLoS Genet 2013, 9:e1004035). To exclude the possibility that tagging with fluorescent
protein affects the phase separation capacity of VPS proteins, we expressed and purified
unlabelled VPS23, VPS28 and VPS37 and found that consistent with tagged proteins, none
of these formed visible condensates in vitro (Response Fig. 9a). In vivo partitioning was
performed according to Gao et al. (Curr Biol 2014, 24:2556-63). VPS23-mCerulean was co-
expressed with GFP-FREE1 in protoplasts.

Condensing and non-condensing FREE1 were always tagged with N-terminal GFP,
which is able to phase separate and fully complements lethal *free1* phenotypes (Fig. 1f). To
further alleviate any concern that tagging affects FREE1 phase separation, we tested phase
separation of the untagged FREE1 protein, finding that condensate formation and fusion
capability were comparable to those of GFP-FREE1 (Response Fig. 9b,c). In the revised
manuscript, we include these supporting data and have also updated the methods and figure
captions.

5. For the mathematical model, it would be very useful to clearly specify the sizes considered
in the simulations. What sizes of ILVs are considered? Figure 3 mentions a radius for MVBs
of 35nm but according to Extended Data Figure 7a, that is the full diameter of a typical ILV.

Response: We apologize for this oversight on our part. In the revised version, we have
re-run all simulations using an enhanced 3D modelling strategy and in silico simulations that
account for condensate sizes corresponding to empirically-determined ILV sizes. We have
updated the new Fig. 4a to show the results of our theoretical scission model.

6. In a previous publication (Gao et al, 2014), it was shown that the FYVE domain of FREE1
was necessary for binding to membranes. The FYVE domain is located outside the IDR. In
this manuscript, the authors show that a FREE1 version without its IDR but with its FYVE
domain loses its capability to bind membranes. Can the authors reconcile these results?

Response: We regret not including a non-membrane-binding negative control in the
membrane floatation assay of our original submission. Compared to non-membrane-binding

GFP, IDR deletion significantly reduced but did not abolish FREE1 membrane association
(revised Fig. 2d). This result suggests that condensation enhances FREE1 membrane
binding capacity provided by the FYVE domain. To support this finding, we tested whether
the presence of the IDR affects PI3P binding by the FYVE domain using a dot blot assay. In
these experiments, we prepared increasing PI3P concentrations on a PVDF membrane
which was incubated with FREE1 or FREE1^{ΔIDR}. Our results show that FREE1 and
FREE1^{ΔIDR} exhibit comparable PI3P binding capacity (Response Fig. 6b) and suggest that
IDR deletion alone does not affect the PI3P binding capacity of the FYVE domain.

To test this in vivo, we also expressed FREE1 in the yeast *Saccharomyces cerevisiae*.
The FYVE domain is known to bind the PI3P-rich vacuolar membrane in yeast. We found
that both FREE1 and FREE1^{ΔIDR} strongly bound the vacuolar membrane when the FREE1
protein contained the FYVE domain (Response Fig. 10). Removal of the FYVE domain
resulted in cytosolic formation of condensates (FREE1) or diffuse cytosolic localisation
(FREE1^{ΔIDR}) (Response Fig. 10), providing very strong evidence that the FYVE domain, not
the IDR, is responsible for membrane binding. We have included the new data and rephrased
the text in our revised manuscript.

7. Related to this topic, the authors use wortmannin to increase the size of MVBs in
Arabidopsis to better visualize the association between FREE1 and MVB membranes.
However, wortmannin blocks the synthesis of PI3P and MVBs under these conditions should
be largely devoid of PI3P. How do the authors explain that still FREE1 associate with MVBs?
In other words, what would be providing the specificity of FREE1 to associate with endosomal
and autophagosome membranes if PI3P is not required for its membrane association?

Response: Thank you for raising this important point. To address this question, we
carried out lipidomic analysis and found that wortmannin treatment reduced cellular PI3P
levels by a factor of five in our MVB enlargement experiments (Response Fig. 6c,d). Next,
we repeated the floatation assay with an up to ten-fold reduction in PI3P concentrations and
observed that the membrane fraction still enriches condensed FREE1 (Response Fig. 6e),
indicating that condensation lowers the requirement of PI3P for FREE1 association with
membrane. Additionally, it has been shown in plants that wortmannin induces homotypic
fusion of MVBs, with the lipid originating from old MVBs (Wang et al., J Exp Bot 2009,
60:3075-83), implying that wortmannin-enlarged MVBs maintain proper PI3P level. Together,
these results explain why FREE1 condensates associate with MVBs following wortmannin
treatment.

8. In line 231, the authors support their claim that FREE1 is internalized into ILVs by showing
FREE1-positive puncta inside Con-A-treated vacuoles (Extended Data Fig 7d). At this
resolution, individual ILVs should not be discernable. As shown before and mentioned in this

manuscript, FREE1 is also transported to the vacuole for degradation by autophagy. The
puncta shown in panel d are most likely autophagic bodies, not ILVs.

Response: We thank the referee for pointing this out. We accept that these FREE1-
positive puncta could be of autophagic origin and have therefore removed these data and
performed a number of additional experiments to investigate FREE1 function in ILV formation.
We overexpressed GFP-IDR^{FUS}-FREE1 in the *vps2.1* mutant (in which the essential ESCRT-
III subunit VPS2 is knocked out) and performed immuno-electron microscopy using gold
particles against GFP. We observed gold particles inside the MVB lumen (Response Fig. 1d).
Consistent with this result, FREE1 condensates enter the lumen of wortmannin-induced large
MVBs (Response Fig. 6a). Further, Xia et al. have previously reported that FREE1 puncta
are observed in the lumen of MVBs and can be degraded via the MVB-vacuolar pathway (Xia
et al., Plant Cell 2020, 32:3290-3310).

Response Fig. 1 | FREE1 condensation compensates for ESCRT machinery defects.

a, Representative images of *A. thaliana* embryos at indicated stages. Scale bars, 50 μm .

b, Statistical analyses of cotyledon lengths and percentage of embryos arrested at different stages.

c, Left, TEM images of MVBs in early embryos of indicated genotypes. Scale bars, 100 nm. Right, quantification

of the number of ILVs per MVB. Error bars indicate mean \pm SD.

d, Immuno-gold labelling of GFP-IDR^{FUS}-FREE1 in *A. thaliana vps2.1* knock-out embryo cells. Scale bar, 50 nm

e, Electrophoresis of the PCR genotyping products using primers as indicated. (Data in (a-c) were included as Fig. 4c-e;

(d) as Extended Fig. 9d; and (e) as Extended Fig. 3f in the revised manuscript)

Response Fig. 2 | FREE1 forms condensates in mammalian COS-7 cells.

a, Full length FREE1 but not FREE1^{ΔIDR} form condensates following transient expression in COS-7

cells. Representative confocal images. Scale bars, 5 μ m. **b**, Representative electron microscopy

images of immunogold-labelled samples using GFP antibodies in COS-7 cells expressing GFP-

FREE1. Scale bar, 100 nm. **c**, Left, mammalian COS-7 cells expressing indicated proteins.

Representative transmission electron microscopy (TEM) images. Right, ILV quantification. Mean \pm

SD. Asterisks indicate significant differences (two-tailed t tests). **d**, Mammalian COS-7 cells co-

expressing full-length FREE1 or FREE1^{ΔIDR} with Rab5^{Q79L} were fixed and imaged by super resolution

(structured illumination) microscopy. Scale bars, 1 μ m. (Data in (a) is included as Extended Data

Figure 2f; (b) as Extended Data Figure 9a; (c) as Fig. 3a and (d) as Extended Data Figure 9b in the

revised manuscript)

Response Fig. 3 | Phase-separated FREE1 condensates partition VPS23.

Left, confocal microscopy images of indicated protoplast cells co-expressing VPS23 with RHA1. Right,

quantification of the colocalization coefficient between VPS23 and RHA1. Data are presented as the

mean ± SD (n = 3 biological replicates). Asterisk indicates significant difference (two-tailed *t* tests).

Scale bars, 5 μm. Data included as Extended Data Fig. 5e in the revised manuscript.

**Response Fig. 4 | Computer simulations of condensate-membrane wetting and ILV formation.**

**a**, Setup of numerical axisymmetric 3D simulations to determine ILV formation dynamics. Initial

bending energy minimizing MVB and GUV shapes were determined by running condensate-free

simulations. **b**, Dynamics of condensate-mediated ILV formation in MVBs (magenta lines) and GUVs

(green lines) for varying ambient viscosities ($\eta_{\text{ambient}} = 0.01\text{-}10$ Pas). Insets show intermediate 3D
morphologies as indicated. Scaled free condensate areas (ratio of condensate/fluid surface with
respect to initial value) decrease over time. $\theta = 70^\circ$, membrane tension $\sigma_{\alpha\gamma} = 1.5$ mN/m, membrane
tension $\sigma_{\alpha\beta} =$ surface tension $\sigma_{\beta\gamma} = 1.1$ mN/m. Condensate viscosity $\eta_{\text{cond}} = 10$ Pas. Condensate
diameters are $2\ \mu\text{m}$ and $35\ \text{nm}$ in the GUV-like and MVB-like simulations, respectively. **c**, Dynamics
of ILV formation for varying ambient viscosities. Scaled free condensate area (ratio of
condensate/fluid surface with respect to initial value) decreases over time. $\theta = 70^\circ$, membrane tension
$\sigma_{\alpha\gamma} = 1.5$ mN/m, membrane tension $\sigma_{\alpha\beta} =$ surface tension $\sigma_{\beta\gamma} = 1.1$ mN/m. Viscosities $\eta_{\text{cond}} = 10$ Pas,
$\eta_{\text{ambient}} = 0.01\text{-}10$ Pas. Condensate diameters, $D_{\text{GUV}} = 2\ \mu\text{m}$, $D_{\text{MVB}} = 35\ \text{nm}$. **d**, 3D Simulation snapshots
comparing membrane (green) and condensate (magenta) intermediate shapes at free condensate
area = 0.5 and $\eta_{\text{cond}}/\eta_{\text{ambient}} = 1$ (darkest) to $\eta_{\text{cond}}/\eta_{\text{ambient}} = 1000$ (lightest 1000). **e**, Stationary 3D shapes
for wetting simulations of large, single condensates ($2\ \mu\text{m}$) and GUVs with increasing values of the
parameter $C = \sigma_{\beta\gamma} D_{\text{ILV}}^2 / \kappa$. **f**, MVB-ILV formation *in silico* time series for conditions as in (c) and line
tension $\lambda = 10$ pN. (Data in (a,c,d,f) are included as Fig. 3d-g; (b,e) are included as Extended Data
Figure 8d,e in the revised manuscript)

Response Fig. 5 | Theoretical stability diagrams of condensate-induced membrane scission.

a, Higher order terms stabilize necks at a finite radius. Energy as a function of neck size and higher order terms being present (solid lines) and absent (dashed lines). As a concrete example we use

$E(R_{ne}) = A \left(\frac{d}{R_{ne}} \right)^2 + f R_{ne}$, where d would be a length scale equivalent to membrane thickness, while

the energy unit and force scale of A would be defined as $f_0 \equiv A/d$. As we can see by comparing

the blue and red solid curves, an increase in constriction force f still acts antagonistically with regard
to higher order corrections, resulting in further squeezing of the neck. The radius of the neck
corresponds to the location of the energy minimum, marked by the arrows, which shifts to smaller
R_{ne} with increasing force. **b**, Theoretical stability diagram of condensate-induced membrane scission
and variable ILV size. Blue line, critical scission line tension λ^* . Critical neck constriction force $f^* = 25$
pN, bending rigidity $\kappa = 10^{-19}$ J, $D_{MVB} = 200$ nm. Grey dashed line indicates $D_{ILV} = 35$ nm. **c**, Theoretical
stability diagram of condensate-induced membrane scission for diverging MVB size. Blue line, critical
scission line tension λ^* . Critical neck constriction force $f^* = 25$ pN, bending rigidity $\kappa = 10^{-19}$ J, $D_{ILV} =$
35 nm. Grey dashed line indicates $D_{MVB} = 200$ nm. **d**, Theoretical stability diagram of condensate-
induced membrane scission and varying membrane asymmetry (or membrane spontaneous
curvature). Blue line, critical scission line tension λ^* . Critical neck constriction force $f^* = 25$ pN, bending
rigidity $\kappa = 10^{-19}$ J, $D_{MVB} = 200$ nm, $D_{ILV} = 35$ nm. **e**, Dynamics of ILV formation for varying line tensions.
Scaled free condensate area (ratio of condensate/fluid surface with respect to initial value) decreases
over time. $\theta = 70^\circ$, membrane tension $\sigma_{\alpha\gamma} = 1.5$ mN/m, membrane tension $\sigma_{\alpha\beta} =$ surface tension $\sigma_{\beta\gamma} =$
1.1 mN/m. Viscosities $\eta_{cond} = 10$ Pas, $\eta_{ambient} = 0.01-10$ Pas. Condensate diameter $D_{MVB} = 35$ nm. **f-h**,
The line tensions required for scission for inward budding (solid line) and outward budding (dashed
line) at three different physiological relevant condensate sizes (30 nm, 300 nm and 3000 nm) and a
range of relative membrane sizes.

(Data in this figure are included as Fig. 4a and Extended Data Figure 10 in the revised manuscript)

**Response Fig. 6 | FREE1 condensates localize to PI3P containing membranes *in vivo* and *in***
 ***vitro*.**

**a**, Confocal microscopy image showing the localization of FREE1 within wortmannin-enlarged MVBs
 in *A. thaliana* root tip cells. Scale bar, 1 μ m. **b**, Dot blot assay showing the binding of condensing and
 non-condensing FREE1 to PI3P. **c**, Ion chromatogram (XIC) of PI3P peak at Rt = 4.79 min, separated
 from PI4P or PI5P peak at Rt = 5.67 min obtained from injecting indicated lipid extract (details of
 sample preparation is discussed in methods). **d**, Normalized PI3P levels; Wortmannin treatment
 reduced PI3P levels to $18.8\% \pm 1.4\%$ ($n=3$). **e**, Immunoblot of total (T) and membrane-bound (MB)
 proteins following membrane flotation assay as in Fig. 2. TEV protease cleavage of the MBP
 solubility tag induces LLPS. (Data in this figure are included as Extended Data Figure 4b-g in the
 revised manuscript)

**Response Fig. 7 | Characterization of IDR^{FLOE1} and IDR^{FUS} used for functionally replacing**
 **IDR^{FREE1}.**

**a**, Percentage of different amino acids classes making up IDRs considered in this study. **b**, Disorder
 of the prion-like domain predicted by the PLAAC algorithm. **c**, In vitro phase separation assay using
 10 μ M of the GFP-labelled FREE1 variants. **d**, Top, seven-day-old *A. thaliana* seedlings of indicated
 genotypes; Bottom, electrophoresis of indicated genotyping products. (Data in this figure are included
 as Extended Data Figure 2h-j and Extended Data Figure 3b,e in the revised manuscript)

**Response Fig. 8 | FREE1 is conserved in land plants.**

**a**, Phylogenetic tree of FREE1 homologs from indicated species. **b**, Domain structures of FREE1
homologs. IDRs were predicted using PONDR. **c**, Multiple sequence alignment of FREE1 homologs
from indicated species. **d**, A table showing the enrichment of FREE1 homologs by b-isox. (Data in
this figure are included as Extended Data Figure 7 in the revised manuscript)

Response Fig. 9 | Phase separation of unlabelled FREE1 and VPS in vitro.

a, Confocal microscopy images of unlabelled VPS proteins at 5 μM concentration. **b**, Confocal microscopy images of FREE1 condensates formed at indicated protein and salt concentrations. **c**,

Coalescence of FREE1 condensates. Scale bars, 5 μm. (Data in this figure are included as Extended

Data Figure 2c, d in the revised manuscript)

Response Fig. 10 | The FREE1 FYVE domain mediates vacuolar membrane localization in *Saccharomyces cerevisiae*.

Indicated variants of GFP-FREE1 (green) were expressed under the control of the GAP1 promoter and observed by confocal fluorescence microscopy (upper panels). Prior to visualization, yeast vacuolar membranes were stained with FM4-64 dye (magenta). Scale bars, 1 μ m. Lower panels, corresponding brightfield images. (This figure is included as Extended Data Figure 5 in the revised manuscript)

Reviewer Reports on the First Revision:

Referees' comments:

Referee #1 (Remarks to the Author):

The revision by Wang et al addresses my questions regarding the experimental science of the work, as well as my most significant technical criticism of the theory (that the shapes observed by the dynamic model did not look right, and that the details of dimensionality were not clearly explained). Conceptually, I would describe the purpose of the theoretical section of the paper as purely supplemental — providing theoretical context that does not strongly affect the story of the work. I am conflicted on the point of the relevance of the dynamic model: I enjoy seeing the model and knowing that the project has been guided and supported in part by this (very cool) theoretical work, but the dynamic predictions do not seem to be testable and the proof of principle does not appear to be critical. I offer this as advice to the editors, that the dynamic model could be moved to supplemental without changing the scientific validity of the main story.

In the words of the authors rebuttal, addressing two reviewers pointing out that the purpose of the dynamical model wasn't clear: "As a result, modelling is the only available strategy to understand wetting dynamics and budding intermediate morphologies for both the GUVs and MVBs." Yet wetting dynamics and intermediate morphologies do not appear to be important for the story.

The Helfrich-based theory does provide some context but is somewhat imbalanced by the number of unknown parameters. In the end the authors use very little real data to feedback into the main text. Essentially: Steinkuhler (2020) found a critical force of 25 pN. If that is the required force, and given the geometry of our system, our line tension must be at least 11.3 pN. This is minor but important modeling effort — taking a (partially) experimentally determined value for the scission force and translating it into the line tension. My minor revision request is that the authors not speak so definitively about the Steinkuhler critical force. From my reading, Steinkuhler found scission in three samples, and 26 pN was the lowest value they observed. They did not map out the critical force systematically. In that sense, making clear in the Figure 4 caption, as well as the main text, the uncertainty in the Steinkuhler value is necessary (unless they can justify that I am wrong about its precision).

I agree with the authors' reply to Reviewer #2's point regarding the phase diagram. The absolute key is a line tension, which is justified theoretically. Yet I do believe the heavy reliance on the (in my reading) imprecise Steinkuhler value for the critical scission force is reckless.

Reviewer #2 and I agreed that the point of the dynamic model was not clear, and was not clear even reading the rebuttal. Nevertheless I enjoyed very much reading their scientific endeavors to validate the model in the rebuttal! I am still enthusiastic about the model generally.

Referee #1 (Remarks on code availability):

The code appears complete and in a format that can be compiled. I could not compile the code on either my Mac or my Centos 7 system for lack of the "DUNE" numerical system.

Referee #3 (Remarks to the Author):

I appreciate the responses of the authors to my queries, which were detailed and appropriate. I particularly appreciate responses to point 3 and associated experiments, which provide important support to the model the authors present. Based on these responses, I recommend publication of this manuscript in Nature.

Referee #4 (Remarks to the Author):

In this revised version, the authors have addressed all my major questions.

I still think they should consider that plant MVBs have been reported to undergo limited ILV released (see Buono et al, JCB, 2017). Therefore, the participation of FREE1 may be critical for membrane deformation but not necessarily to complete ILV scission in vivo. It would be appropriate to consider this in the manuscript since lack/limited membrane fission in MVBs seems to be a unique features of plants, just like the presence of FREE1.

Author Rebuttals to First Revision:

Referee #1:

1. The revision by Wang et al addresses my questions regarding the experimental science of the work, as well as my most significant technical criticism of the theory (that the shapes observed by the dynamic model did not look right, and that the details of dimensionality were not clearly explained).

Response: We thank the reviewers for their positive assessment of our work and appreciation of the theoretical approaches developed.

2. Conceptually, I would describe the purpose of the theoretical section of the paper as purely supplemental — providing theoretical context that does not strongly affect the story of the work. I am conflicted on the point of the relevance of the dynamic model: I enjoy seeing the model and knowing that the project has been guided and supported in part by this (very cool) theoretical work, but the dynamic predictions do not seem to be testable and the proof of principle does not appear to be critical. I offer this as advice to the editors, that the dynamic model could be moved to supplemental without changing the scientific validity of the main story.

In the words of the authors rebuttal, addressing two reviewers pointing out that the purpose of the dynamical model wasn't clear: "As a result, modelling is the only available strategy to understand wetting dynamics and budding intermediate morphologies for both the GUVs and MVBs." Yet wetting dynamics and intermediate morphologies do not appear to be important for the story.

The Helfrich-based theory does provide some context but is somewhat imbalanced by the number of unknown parameters. In the end the authors use very little real data to feedback into the main text. Essentially: Steinkuhler (2020) found a critical force of 25 pN. If that is the required force, and given the geometry of our system, our line tension must be at least 11.3 pN. This is minor but important modeling effort — taking a (partially) experimentally determined value for the scission force and translating it into the line tension. My minor revision request is that the authors not speak so definitively about the Steinkuhler critical force. From my reading, Steinkuhler found scission in three samples, and 26 pN was the lowest value they observed. They did not map out the critical force systematically. In that sense, making clear in the Figure 4 caption, as well as the main text, the uncertainty in the Steinkuhler value is necessary (unless they can justify that I am wrong about its precision).

I agree with the authors' reply to Reviewer #2's point regarding the phase diagram. The absolute key is a line tension, which is justified theoretically. Yet I do believe the heavy reliance on the (in my reading) imprecise Steinkuhler value for the critical scission force is reckless. Reviewer #2 and I agreed that the point of the dynamic model was not clear, and was not clear even reading the rebuttal. Nevertheless I enjoyed very much reading their scientific endeavors to validate the model in the rebuttal! I am still enthusiastic about the model generally.

Response: In the revised version, we have included a comment that the Steinkuehler critical force should be taken as an order of magnitude estimate. Further, we considered the reviewer's suggestion regarding the dynamical model and moved parts of it to supplemental. We thank the reviewer for helping us to improve the clarity of our work.

3. The code appears complete and in a format that can be compiled. I could not compile the code on either my Mac or my Centos 7 system for lack of the "DUNE" numerical system.

Response: We appreciate the reviewers ambition and interest to install our code. The reviewer might have overlooked that we share detailed instructions (about 4 pages) how to install the code including DUNE. These instructions can be found on the given Zenodo website (in README.md) and on the starting page of the linked gitlab repository. There, we also pointed out clearly that the code cannot be compiled on a Mac system. Please also note that installation of such state-of-the-art finite-element code will take several hours and might require good knowledge of Linux.

Referee #3:

I appreciate the responses of the authors to my queries, which were detailed and appropriate. I particularly appreciate responses to point 3 and associated experiments, which provide important support to the model the authors present. Based on these responses, I recommend publication of this manuscript in Nature.

Response: We thank reviewers for their support and positive recommendation.

Referee #4:

In this revised version, the authors have addressed all my major questions. I still think they should consider that plant MVBs have been reported to undergo limited ILV released (see Buono et al, JCB, 2017). Therefore, the participation of FREE1 may be critical for membrane deformation but not necessarily to complete ILV scission in vivo. It would be appropriate to consider this in the manuscript since lack/limited membrane fission in MVBs seems to be a unique features of plants, just like the presence of FREE1.

Response: We thank the reviewer the positive feedback. We note the reviewer's suggestion to refer to Buono et al. (2017), which we now cite. However, we would like to briefly justify our decision not to cite it specifically as suggested by the reviewer:

- The suggested paper shows that the number of incompletely scissioned ILVs increases with increasing ILV number, which could be caused by a depletion of scission inducing machinery/condensates. We therefore respectfully disagree with the reviewer's interpretation of this work.

- Meanwhile, the paper shows that ESCRT machinery deletions (chmp1, lip5) decreases the number of ILVs but increases the fraction of free ILVs. This suggests the existence of a machinery-independent scission mechanism, which we feel is relevant background to our description of the mechanism governing such scission.

- Further, the paper predicts hinderance of free diffusion in ILVs, which could be due to the formation of condensates within ILVs.

We therefore have decided to refer to this paper at line 390 as below:
“Condensates also could provide a means to hinder free diffusion inside ILVs [Buono 2017]”